# 21st century fate of the Mocho-Choshuenco ice cap in southern Chile

Matthias Scheiter[1,2], Marius Schaefer[3], Eduardo Flández[4], Deniz Bozkurt[5,6], and Ralf Greve[7,8]

[1]Research School of Earth Sciences, Australian National University, Canberra, Australia
[2]previously at Institut für Geophysik und Geoinformatik, TU Bergakademie Freiberg, Freiberg, Germany
[3]Instituto de Ciencias Físicas y Matemáticas, Universidad Austral de Chile, Valdivia, Chile
[4]Departamento de Física, Facultad de Ciencias, Universidad de Chile, Santiago, Chile
[5]Departamento de Meteorología, Universidad de Valparaíso, Valparaíso, Chile
[6]Center for Climate and Resilience Research (CR)2, Santiago, Chile
[7]Institute of Low Temperature Science, Hokkaido University, Sapporo, Japan
[8]Arctic Research Center, Hokkaido University, Sapporo, Japan

**Correspondence:** Matthias Scheiter (matthias.scheiter@anu.edu.au)

**Abstract.** Glaciers and ice caps are thinning and retreating along the entire Andes ridge, and drivers of this mass loss vary between the different climate zones. The southern part of the Andes (Wet Andes) has the highest abundance of glaciers in number and size, and a proper understanding of ice dynamics is important to assess their evolution. In this contribution, we apply the ice sheet model SICOPOLIS to the Mocho-Choshuenco ice cap in the Chilean Lake District (40°S, 72°W, Wet Andes) to reproduce its current state and to project its evolution until the end of the 21st century under different global warming scenarios. First, we create a model spin-up using observed surface mass balance data on the south-eastern catchment, extrapolating them to the whole ice cap using an aspect-dependent parameterization. This spin-up is able to reproduce the most important present-day glacier features. Based on the spin-up, we then run the model 80 years into the future, forced by projected surface temperature anomalies from different global climate models under different radiative pathway scenarios to obtain estimates of the ice cap's state by the end of the 21st century. The mean projected ice volume losses are $56 \pm 16\%$ (RCP2.6), $81 \pm 6\%$ (RCP4.5) and $97 \pm 2\%$ (RCP8.5) with respect to the ice volume estimated by radio-echo sounding data from 2013. We estimate the uncertainty of our projections based on the spread of the results when forcing with different global climate models and on the uncertainty associated with the variation of the equilibrium line altitude with temperature change. Considering our results, we project a considerable deglaciation of the Chilean Lake District by the end of the 21st century.

## 1 Introduction

Most glaciers and ice caps in the Andes are currently thinning and retreating (e.g. Braun et al., 2019), and rates of mass loss are increasing in many places (Dussaillant et al., 2019). In the Southern part of the Andes (Wet Andes or Patagonian Andes, 36-56°S) the highest number of glaciers are found and large icefields such as the Northern Patagonia Icefield, Southern Patagonia

Icefield and Cordillera Darwin are located in this region. The specific mass losses observed or inferred for the glaciers of the Wet Andes are the highest in the Andes (Dussaillant et al., 2019; Braun et al., 2019) and among the highest of all glacier regions worldwide (Zemp et al., 2019).

The maritime climate of the Wet Andes is characterized by high precipitation rates of up to $10\,\mathrm{m\,yr^{-1}}$ on the windward side and rather mild temperatures with freezing levels generally above 1 km above mean sea level, with an overall modest seasonality (Garreaud et al., 2013). This leads to an exceptionally high mass turnover (Schaefer et al., 2013, 2015, 2017) and high flow speeds for the glaciers in the region (Sakakibara and Sugiyama, 2014; Mouginot and Rignot, 2015). In addition to climate forcings, other important contributors to glacier change in the region are ice dynamics and frontal ablation. Ice-flow models incorporate these processes, and are therefore appropriate tools to project the future behaviour of the glaciers of the Wet Andes.

Only few studies have tried to project future behaviour of Andean glaciers. Réveillet et al. (2015) modelled Zongo Glacier (16°S) in the tropical Andes using the 3-D full-Stokes model Elmer/Ice (developed by Gagliardini et al., 2013). They projected volume losses between 40% and 89% until the end of this century under the RCP2.6 and RCP8.5 scenarios, respectively. In the Wet Andes, Möller and Schneider (2010) projected an area loss of 35% of Glaciar Noroeste, an outlet glacier of the Gran Campo Nevado ice cap (53°S), until the end of the 21st century using a degree-day model and volume-area scaling relationships. Schaefer et al. (2013) modeled the surface mass balance (SMB) of the Northern Patagonian Icefield in the 21st century under the A1B scenario (of IPCC Assessment Report 4, comparable to RCP6.0). They projected a strongly decreasing SMB until the end of the 21st century, mainly due to an increase in surface temperature by the middle of the century and a decrease of accumulation towards the end of the century. Collao-Barrios et al. (2018) infer important committed mass loss of San Rafael Glacier under current climate applying the Elmer/Ice flow model with fixed glacier outlines.

In this contribution, our first objective is to reproduce the present-day behaviour of the Mocho-Choshuenco ice cap in the northern part of the Wet Andes (40°S) using the ice-sheet model SICOPOLIS (Greve, 1997a, b). To this end, we make use of a newly developed SMB parameterization scheme and glaciological data obtained on the ice cap to calibrate the model and reproduce its current state. Our second objective is to project the behaviour of the Mocho-Choshuenco ice cap through the course of the 21st century to provide one of the first constraints on future glacier dynamics in the Wet Andes. For this aim, we make use of temperature projections from 23 Global Climate Models (GCMs) participating in the Coupled Model Intercomparison Project phase 5 (CMIP5) (Taylor et al., 2012) under low (RCP2.6), medium (RCP4.5) and high emission (RCP8.5) scenarios as input to SICOPOLIS.

We begin this paper by describing the observational data and methods (section 2). In section 3, we present the results: first, we validate the model spin-up using observed SMB, glacier outlines, ice thickness and flow speed. We then present the evolution of ice cap extension and volume during the 21st century as obtained through different emission scenarios. Then, in section 4, we discuss our results, compare them to previous studies and analyse the limitations of our approach. We conclude the paper by summarizing the main findings in section 5.

## 2 Methods

### 2.1 Observational data

The ice cap on which we focus in this study covers the Mocho-Choshuenco volcanic complex, which is located at $40°$S, $72°$W (see inset map in Figure 1). Over the last 20 years, climatological and glaciological observations have been made on the ice cap (Rivera et al., 2005; Schaefer et al., 2017). SMB data were obtained through the traditional glaciological method on a stake network on the south-eastern part of the ice cap (red stars in Figure 1). These measurements reported by Schaefer et al. (2017) yielded an average negative SMB of $-0.9\,\mathrm{m\,w.e.\,yr}^{-1}$ (meter water equivalent per year) with a high mass turnover of around $2.6\,\mathrm{m\,w.e.\,yr}^{-1}$ (see section 2.4). This high mass turnover is a consequence of the interaction between high precipitation rates leading to high accumulation rates, and high temperatures leading to high melt rates. In this respect, climatological data (2006 to 2015) indicate that the annual mean temperature was $2.6°$C at an automatic weather station (green circle in Figure 1) at an elevation of $2000\,m$, and therefore, close to the typical equilibrium line altitude (ELA) (Schaefer et al., 2017). Mean annual precipitation over the same period was around $4000\,\mathrm{mm\,yr}^{-1}$ in Puerto Fuy at an elevation of $600\,m$ to the north of the volcano, and orographic precipitation effects lead to a relatively high amount of precipitation on the ice cap.

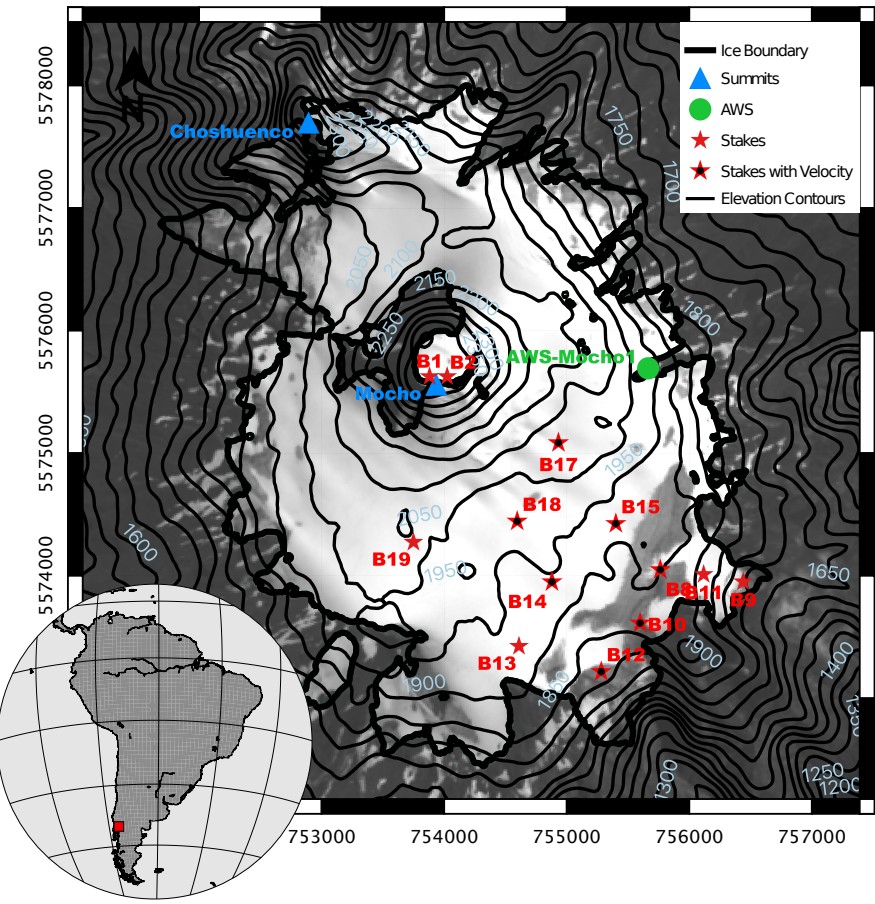

**Figure 1.** Overview map of the Mocho-Choshuenco ice cap with significant geographic features and measurement sites. The contour line spacing is 50 m. East and North are in UTM S18. Background: Landsat image (February 22, 2015). Inset map shows location in South America.

At some of the mass balance stakes (red stars with inner black dots in Figure 1), high precision GPS measurements were made in July and October 2013 to infer surface flow velocity (Geoestudios, 2013), and the results are shown in Table 1. Further measurements include ground penetrating radar (GPR) transects (green lines in Figure 2a) over most parts of the ice cap (Geoestudios, 2014). Through inverse distance weighting interpolation over the whole ice cap, a total ice volume of $1.038\,\mathrm{km}^3$ was obtained (Geoestudios, 2014). The interpolated ice thickness map was subtracted from a digital elevation model (TanDEM WorldDEM, acquired between 2012 and 2014) to yield a bedrock topography (Flández, 2017). We use this topography as the base of the ice cap in the simulations we perform with SICOPOLIS.

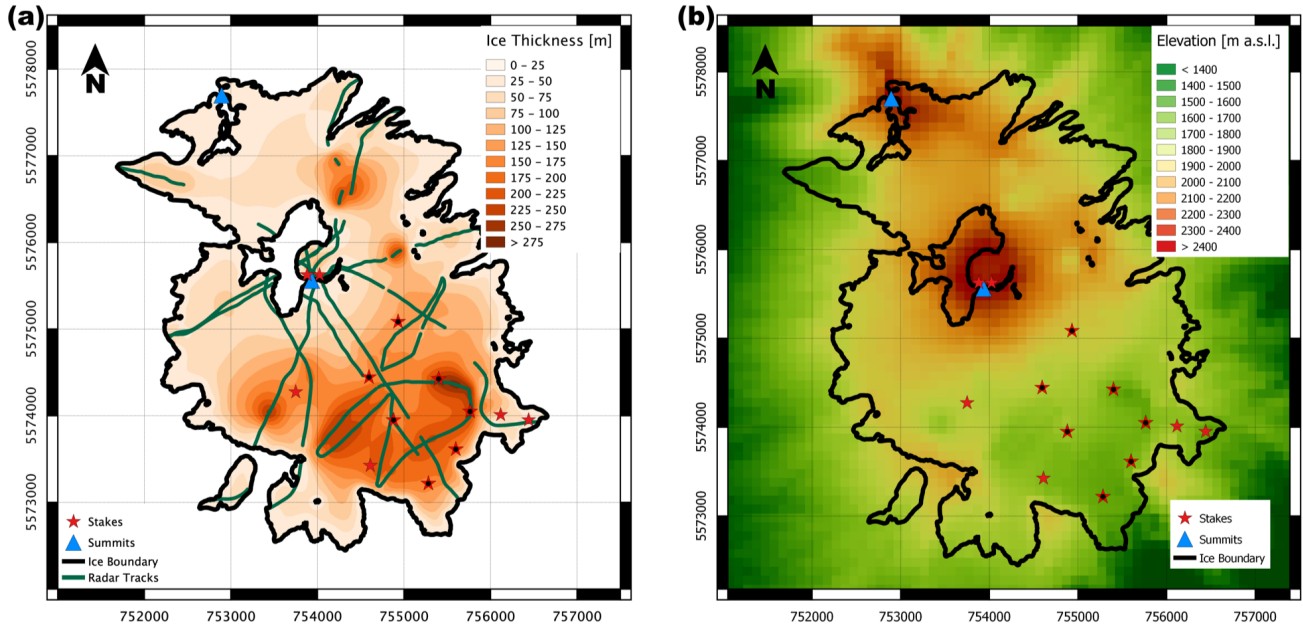

**Figure 2.** (a) Ground penetrating radar (GPR) transects shown in green lines together with the interpolated ice thickness. (b) Bedrock topography obtained after subtracting the interpolated ice thickness from surface elevation. This topography is used as the ice cap base in our simulations.

## 2.2  SICOPOLIS

The three-dimensional, dynamic/thermodynamic model SICOPOLIS (SImulation COde for POLythermal Ice Sheets) was originally created in a version for the Greenland ice sheet (Greve, 1997a, b). Since then, the model has been developed continuously and applied to problems of past, present and future glaciation of Greenland, Antarctica, the entire northern hemisphere, the polar ice caps of the planet Mars and others, resulting in more than 120 publications in the peer-reviewed literature (www.sicopolis.net). The model supports the shallow-ice approximation (SIA) for slow-flowing grounded ice, hybrid shallow-ice–shelfy stream dynamics for fast-flowing grounded ice and the shallow-shelf approximation for floating ice (Bernales et al., 2017), as well as several thermodynamics solvers (Blatter and Greve, 2015; Greve and Blatter, 2016).

Mainly developed for ice sheets, the smallest ice body to which SICOPOLIS has been applied so far is the Austfonna Ice Cap, for which Dunse et al. (2011) reproduced the observed cyclic surge behaviour under constant, present-day climate conditions. For this study, we adapted SICOPOLIS v5.1 (Greve and SICOPOLIS Developer Team, 2019) for the Mocho-Choshuenco ice cap in SIA mode. We employ a standard Glen flow law with a stress exponent of $n = 3$. Basal sliding is modelled by a linear sliding law,

$$v_{\mathrm{b}} = -C_{\mathrm{b}}\tau_{\mathrm{b}}\,, \tag{1}$$

where $v_b$ is the basal sliding velocity, $\tau_b$ the basal drag and $C_b$ the sliding coefficient. The value of the latter is determined by the calibration procedure of the present-day spin-up (see section 3.1). Since Mocho-Choshuenco is a temperate ice cap, we do not solve the energy balance equation. Rather, we keep the temperature at a constant value of $0°C$ (precisely speaking, and for technical reasons only as SICOPOLIS does not allow an all-temperate ice body, $-0.001°C$). The rate factor is set to the value recommended by Cuffey and Paterson (2010) for $0°C$, which is $A = 2.4 \times 10^{-24}\,\mathrm{s}^{-1}\,\mathrm{Pa}^{-3}$. To ensure proper mass conservation despite the steep slopes and rugged bed topography, we use an explicit solver for the ice thickness equation that discretizes the advection term by a mass-conserving scheme in an upwind flux form (Calov et al., 2018).

## 2.3 Aspect-dependent SMB parameterization

SICOPOLIS incorporates a linear altitude-dependent SMB parameterization which is visualised in Figure 3a and can be described by the following formula:

$$\text{SMB}(C) = \min(S_0, M_0 \cdot (z(C) - \text{ELA})). \tag{2}$$

Here, ELA is the equilibrium line altitude, $z(C)$ is the evolving ice surface elevation of a specific grid cell $C$, $M_0$ denotes the mass balance gradient and $S_0$ is maximum SMB.

From the simulations performed by Flández (2017) on the Mocho-Choshuenco ice cap it becomes apparent that the simple altitude-dependent SMB parameterization in Equation 2 is not detailed enough to account for small-scale SMB variations on the ice cap. In particular, SMB should be lower in the north-western part than in the south-eastern part of the ice cap, due to the aspect dependence of solar radiation and snow redistribution (wind drift) which during precipitation events predominantly blows from the north-west. We therefore employ a new parameterization which is illustrated in Figure 3b. With the Mocho summit in the center, ELA should have a maximum $B_{\text{ELA}} + A_{\text{ELA}}$ in the direction $\varphi_0$, a minimum $B_{\text{ELA}} - A_{\text{ELA}}$ in the opposite direction and a mean value $B_{\text{ELA}}$ on the two perpendicular directions.

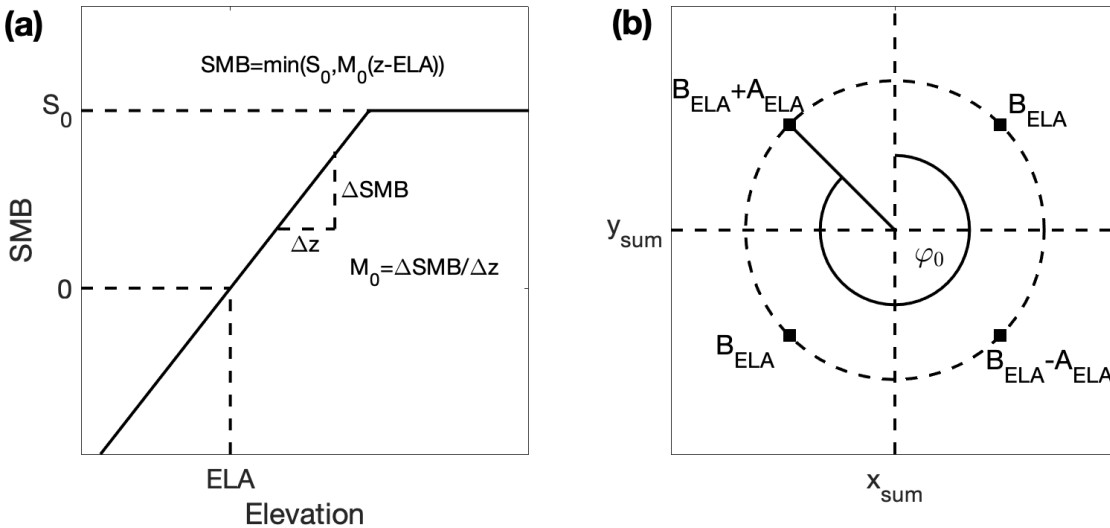

**Figure 3.** (a) Elevation-dependent SMB parameterization. SMB increases linearly with elevation until an upper bound $S_0$ and stays constant at higher elevations. (b) Aspect-dependent SMB parameterization. ELA takes a minimum and maximum on two opposite directions ($B_{\text{ELA}} \pm A_{\text{ELA}}$), and their mean ($B_{\text{ELA}}$) on perpendicular directions. $\varphi_0$ is a direction offset to rotate the values according to the atmospheric conditions. In this visualisation, $\varphi_0$ is set to $315°$, the value used in this study. ($x_{\text{sum}}, y_{\text{sum}}$) indicates the position of Mocho summit.

These values can be summarized in a cosine function in $\varphi$ with the direction of maximum ELA $\varphi_0$, the amplitude $A_{\text{ELA}}$ and an offset of the average ELA $B_{\text{ELA}}$:

$$\text{ELA} = A_{\text{ELA}} \cos(\varphi - \varphi_0) + B_{\text{ELA}}. \tag{3}$$

$B_{\text{ELA}}$ is used to shift the ELA to the desired mean altitude. $\varphi$ is the cardinal direction of a point with respect to the summit and can be calculated by

$$\varphi = \arctan2(x - x_{\text{sum}}, y - y_{\text{sum}}), \tag{4}$$

where arctan2 denotes the two-argument arctangent and $x$ and $y$ are the distances in the two directions from a grid point to the summit location ($x_{\text{sum}}, y_{\text{sum}}$).

**2.4  Transient spin-up**

Before being able to make future projections for the Mocho-Choshuenco ice cap, we first aim to reproduce its current state. Due to the observed negative SMB at present, we aim to build a transient spin-up that represents a shrinking ice cap. This is achieved in two steps: first, we build a theoretical steady-state of the ice cap in the late 1970s, and then run the model from

1979 to 2013 with ERA5 near-surface air temperature data (see Figure 4). This 35-year period is justified by the turnover time

$\tau$, which is a typical time scale for a glacier defined by

$$\tau = \frac{[H]}{[\text{SMB}]}, \tag{5}$$

where $[H]$ is the typical ice thickness and $[\text{SMB}]$ the typical SMB (e.g., Greve and Blatter, 2009). By taking $[H] = V_{\text{obs}}/A_{\text{obs}}$ (where $V_{\text{obs}} = 1.038\,\text{km}^3$ is the observed ice volume and $A_{\text{obs}} = 15.1\,\text{km}^3$ the observed area) and $[\text{SMB}] = 2.6\,\text{m w.e.\,yr}^{-1}$ (computed as the mean of the absolute values from the observed SMB at the stakes), we obtain $\tau \approx 27$ years. This is slightly

less than the 35-year period of ERA5 data, which therefore should be sufficient to produce a valuable spin-up for the year 2013.

ERA5 is a state-of-the-art global reanalysis produced by European Centre for Medium-Range Weather Forecasts (ECMWF). It combines large amounts of historical observations into global estimates using advanced modeling systems and data assimilation, i.e., Integrated Forecasting System (Cycle 41r2) (Hersbach et al., 2020). ERA5 has a spatial resolution of $0.25 \times 0.25$ degree ($\sim 30$ km) and vertical resolution of 137 levels from the surface to a height of 80 km.

Given that there is no available long-term surface meteorological data around the ice cap, we contrasted 700 hPa ERA5 temperature data against the radiosonde data (Integrated Global Radiosonde Archive v2, available at https://www1.ncdc.noaa. gov/pub/data/igra/) from Puerto Montt (41.5°S, 72.9°W) for the period 1979-2019. This is due to the fact that Schaefer et al. (2017) found a very good correlation between the 700 hPa pressure level temperature from the radiosonde data at Puerto Montt and temperature measured at the Mocho automatic weather station. In this respect, ERA5 shows reasonable skills in capturing

the long-term regional temperature trend (+0.19 °C/41 yr) detected in the radiosonde data (+0.22 °C/41 yr) with a high temporal correlation (0.77).

Due to this temperature increase of around 0.2 °C, we build the steady-state by lowering the mean ELA ($B_{\text{ELA}}$) by 18 m in 1979 with respect to the state in 2013, according to the ELA-temperature gradient of $88\,\text{m K}^{-1}$ which we determine in section 2.6. Afterwards, we adjust the model parameters mean ELA ($B_{\text{ELA}}$), ELA amplitude ($A_{\text{ELA}}$), maximum SMB ($S_0$), SMB gradi-

ent ($M_0$), direction of maximum ELA ($\varphi_0$) and sliding coefficient ($C_b$) in order to match the present-day observations of SMB, ice thickness, extent, volume, and surface velocity of the ice cap. While the parameters defining the SMB parameterization were calibrated under observational constraints, $C_b$ was purely used as a calibration parameter. We discuss this in more detail in section 4.1. It is important to note that the steady-state spin-up in the 1970s is a theoretical construct, as the glacier has been losing mass before this period and was not in a steady state. It is only to be interpreted as a first step in order to get an accurate

representation of the shrinking ice cap in 2013 with its negative SMB.

## 2.5   Temperature projections

The main goal of this study is to project the future evolution of the Mocho-Choshuenco ice cap. We use future temperature simulations from 23 climate models participating in CMIP5 (see Appendix A). To ease the calculations, all the models were interpolated onto a common grid of $1.5° \times 1.5°$ using bilinear interpolation. Then the time series of each model were extracted

from the grid point corresponding to Mocho-Choshuenco ice cap (40°S, 72°W). As the model trajectories start in 2006 and in order to be consistent with the reference ice cap conditions based on the observational dataset obtained between 2009 and 2013,

we used the period from 2006 to 2020 as the reference period rather than the commonly used historical periods (e.g., 1976-2005) in order to construct projections of temperature anomalies. For each of the individual models, the mean temperature between 2006 and 2020 was then subtracted from the whole time series, leading to anomaly temperature projections with respect to this period. At the final step, the SICOPOLIS model was driven by each of the 23 model projections to provide a more robust assessment of future evolution of the Mocho-Choshuenco ice cap. This allows us to assess the uncertainty associated with climate model differences. In addition to the future projections, we also include a control run in our analysis where we run the model for the period 2013-2100 with zero temperature anomaly with respect to the reference period 2006-2020. This enables us to calculate a committed mass loss and assess the influence of ice dynamics alone, independent of future temperature increase.

Our approach makes use of three emission scenarios following the IPCC protocols (IPCC, 2013): high-mitigation, Paris Agreement compatible (RCP2.6); medium stabilisation scenario with a peak around 2040, then decline (RCP4.5); and high-end baseline scenario with no control policies of greenhouse gas emissions (RCP8.5). This allows us to contrast the future evolution of the Mocho-Choshuenco ice cap under different emission scenarios together with the uncertainty introduced by future emissions. Figure 4 shows the projected changes in temperature obtained from 23 individual climate models for the ice cap under the three different emission scenarios until the end of the century. This yields 69 projections which are all used to run SICOPOLIS and averaged afterwards. All projections follow a similar trend until the 2040s, where the RCP8.5 scenario separates from the others and continues to increase throughout the century, leading to a model mean temperature increase of $3.15 \pm 0.69°C$ by the end of the century. The temperature projections under the RCP2.6 and RCP4.5 scenarios have largely similar evolutions after the 2050s with weaker projected changes than those in RCP8.5. By the end of the century, projected temperature increases for the RCP2.6 and RCP4.5 scenarios are $0.33 \pm 0.47°C$ and $1.01 \pm 0.43°C$, respectively.

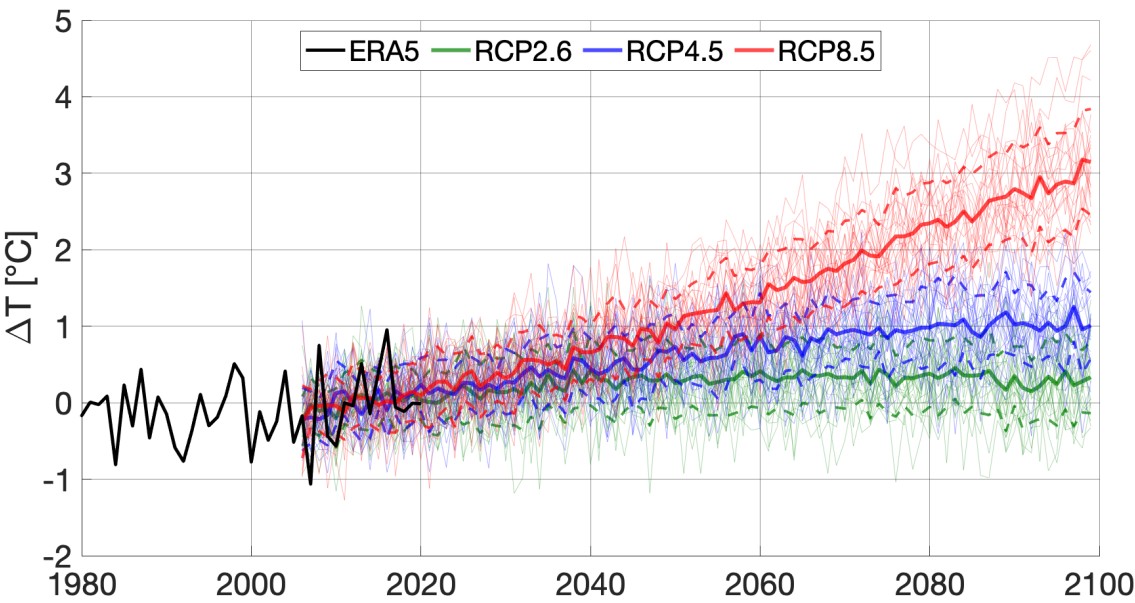

**Figure 4.** Temperature projections for the Mocho-Choshuenco ice cap through the 21st century for three different scenarios RCP2.6, RCP4.5 and RCP8.5, along with historical ERA5 data which is used for the transient spin-up. The thin lines show projections of the 23 individual climate models, thick solid lines indicate their mean and thick dashed lines the one-sigma confidence interval. The period between 2006 and 2020 is used as reference period for each individual model.

## 2.6 Glacier sensitivity to temperature change

To link the projected 21st century temperature rise to ice dynamics, it is necessary to relate the temperature anomalies to changes in SMB which is determined by the mean ELA ($B_{\mathrm{ELA}}$) in our case. We assume that temperature is the only influencing factor on the projected net SMB, without explicitly distinguishing between precipitation and runoff. Further, we focus on annual rather than melt-season temperature projections, as the climate models project both to be very close to each other on the Mocho-Choshuenco volcanic complex. There are four years (2009-2013) where both the ELA and annual mean temperature at a similar altitude are available (Schaefer et al., 2017). These data are shown in Figure 5 together with the ELA error estimates.

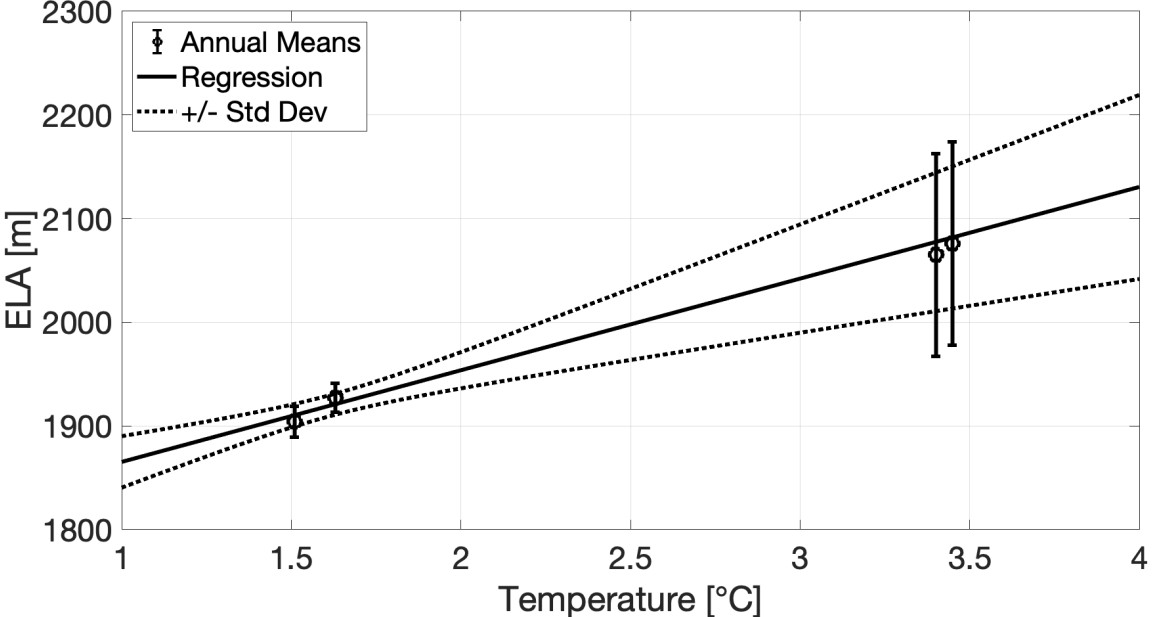

**Figure 5.** Relationship between annual temperature and ELA on Mocho-Choshuenco ice cap. The error bars indicate the error as estimated by Schaefer et al. (2017). The relationship between ELA and temperature was found through weighted linear regression.

In order to predict the ELA for any temperature, we first assume a linear relationship between both and solve a weighted least squares problem to find the slope and intercept (i.e. ELA gradient and ELA for $0°C$). ELA predictions for any temperature $\{T_i, T_j, ...\}$ can be made by multiplying the forward operator $\hat{\mathbf{G}}$ with the vector $\mathbf{m}$ containing both model parameters:

$$\text{ELA} = \hat{\mathbf{G}}\mathbf{m} = \hat{\mathbf{G}}\mathcal{N}(\boldsymbol{\mu}, \boldsymbol{\Sigma}) = \mathcal{N}\left(\hat{\mathbf{G}}\boldsymbol{\mu}, \hat{\mathbf{G}}\boldsymbol{\Sigma}\hat{\mathbf{G}}^T\right), \qquad \hat{\mathbf{G}} = \begin{pmatrix} T_i & 1 \\ T_j & 1 \\ \vdots & \vdots \end{pmatrix}. \tag{6}$$

$\mathbf{m}$ is distributed according to a bivariate normal distribution $\mathcal{N}(\boldsymbol{\mu}, \boldsymbol{\Sigma})$ with mean vector $\boldsymbol{\mu}$ and model covariance matrix $\boldsymbol{\Sigma}$ (e.g. Aster et al., 2018):

$$\mathbf{m} \sim \mathcal{N}(\boldsymbol{\mu}, \boldsymbol{\Sigma}), \qquad \boldsymbol{\mu} = \left(\hat{\mathbf{G}}^T \boldsymbol{\Sigma_d}^{-1} \hat{\mathbf{G}}\right)^{-1} \hat{\mathbf{G}}^T \boldsymbol{\Sigma_d}^{-1} \hat{\mathbf{d}}, \qquad \boldsymbol{\Sigma} = \left(\hat{\mathbf{G}}^T \boldsymbol{\Sigma_d}^{-1} \hat{\mathbf{G}}\right)^{-1}. \tag{7}$$

Inserting the observed data, we identify the forward operator $\hat{\mathbf{G}}$, the data covariance matrix $\boldsymbol{\Sigma_d}$ and the vector of observed ELAs $\hat{\mathbf{d}}$ as

$$
\hat{\mathbf{G}} = \begin{pmatrix} T_1 & 1 \\ T_2 & 1 \\ T_3 & 1 \\ T_4 & 1 \end{pmatrix}, \qquad \boldsymbol{\Sigma_d} = \begin{pmatrix} \sigma_1^2 & 0 & 0 & 0 \\ 0 & \sigma_2^2 & 0 & 0 \\ 0 & 0 & \sigma_3^2 & 0 \\ 0 & 0 & 0 & \sigma_4^2 \end{pmatrix}, \qquad \hat{\mathbf{d}} = \begin{pmatrix} \mathrm{ELA}_1 \\ \mathrm{ELA}_2 \\ \mathrm{ELA}_3 \\ \mathrm{ELA}_4 \end{pmatrix}. \tag{8}
$$

Predictions for a general temperature $T$ can be made through

$$
\mathrm{ELA}(T) = \mathcal{N}\left(\mu_1 T + \mu_2,\ \Sigma_{11} T^2 + 2\Sigma_{12} T + \Sigma_{22}\right), \tag{9}
$$

with

$$
\boldsymbol{\mu} = \begin{pmatrix} 88\,\mathrm{m\,K^{-1}} \\ 1777\,\mathrm{m} \end{pmatrix}, \qquad \boldsymbol{\Sigma} = \begin{pmatrix} 1365\,\mathrm{m^2\,K^{-2}} & -2203\,\mathrm{m^2\,K^{-1}} \\ -2203\,\mathrm{m^2\,K^{-1}} & 3657\,\mathrm{m^2} \end{pmatrix}, \tag{10}
$$

where $\mu_1 = 88\,\mathrm{m\,K^{-1}}$ is our estimated increase of ELA per °C, and $\mu_2 = 1777\,\mathrm{m}$ is the ELA that we would obtain for a yearly average temperature of 0°C. Figure 5 shows the mean of ELA predictions against temperature together with the ELA predictions of mean plus and minus one standard deviation.

Since the temperature projections give anomalies with respect to the period 2006-2020, we only rely on relative rather than absolute temperatures. Therefore, we convert the temperature changes into changes of ELA with the parameter $\mu_1 = 88\,\mathrm{m\,K^{-1}}$, which means that the ELA increases by 88 m per °C temperature increase. We assess the uncertainty propagation of this parameterization through the ice flow simulation code by performing additional experiments with upper and lower ELA gradients $\mu_1 \pm \sqrt{\Sigma_{11}} = (88 \pm 37)\,\mathrm{m\,K^{-1}}$ which corresponds to the one-sigma confidence interval.

## 3 Results

### 3.1 Spin-up and model calibration

Following the spin-up and calibration procedure explained in section 2.4, we tune the model to find the following optimal parameters: $B_{\mathrm{ELA}} = 2050\,\mathrm{m}$, $A_{\mathrm{ELA}} = 87.5\,\mathrm{m}$, $S_0 = 2.2\,\mathrm{m\,a^{-1}}$ and $C_b = 1.0 \times 10^{-4}\,\mathrm{m\,a^{-1}\,Pa^{-1}}$, $M_0 = 0.027\,\mathrm{yr^{-1}}$ and $\varphi_0 = 315°$. We discuss the physical plausibility of these values in section 4.1.

The spin-up is evaluated against observations in Figure 6. Figure 6a shows the thickness distribution and extent of the simulated ice cap. The model captures the general outlines of the ice cap with only small inaccuracies at some outlet tongues. In Figure 6b, we compare the simulated and observed ice thickness. Overall, the simulations overestimate ice thickness in

the northern part of the ice cap, and underestimate it in the south-east. Figure 6c shows that the simulated ice thickness is in reasonable agreement with observations along the radar profiles, with a high correlation (0.91), and the root mean square error (RMSE) that is around 13% of the maximum measured ice thickness.

The velocity map in Figure 6d shows velocities of less than $50 \, \mathrm{m \, yr^{-1}}$ on most parts of the ice cap, matching well with the observed low velocities that were measured in spring 2013 (Geoestudios, 2013). Stakes where velocity measurements are available are marked with black stars in Figure 6d. Observed and modelled velocities at these locations are compared in Table 1, showing an overall good agreement (RMSE of $12.5 \, \mathrm{m \, yr^{-1}}$), with simulated velocities being on average $9.4 \, \mathrm{m \, yr^{-1}}$ lower. However, the modelled velocities represent a yearly average, whereas the velocity measurements were taken in spring season, making a direct comparison difficult, and these values should only be seen as a rough orientation.

| Stake | B8 | B10 | B12 | B14 | B15 | B17 | B18 |
|---|---|---|---|---|---|---|---|
| $v_{\mathrm{obs}}$ $[\mathrm{m \, yr^{-1}}]$ | 22.2 | 12.7 | 60.3 | 33.8 | 19.4 | 31.2 | 27.2 |
| $v_{\mathrm{sim}}$ $[\mathrm{m \, yr^{-1}}]$ | 11.6 | 13.7 | 35.9 | 20.2 | 20.7 | 18.6 | 20.5 |

**Table 1.** Comparison between simulated and observed velocities at stakes where velocity observations are available from Geoestudios (2013).

The simulated SMB in Figure 6e matches well with observations reported by Schaefer et al. (2017) with the observed SMB distribution, SMB gradient and ELA on the south-eastern catchment. Figure 6f shows a direct comparison of modelled SMB at the stake locations and the respective observations. The fit is very good, with a high correlation (0.94), and the RMSE corresponds to roughly 11% of the absolute range between highest and lowest observed SMB.

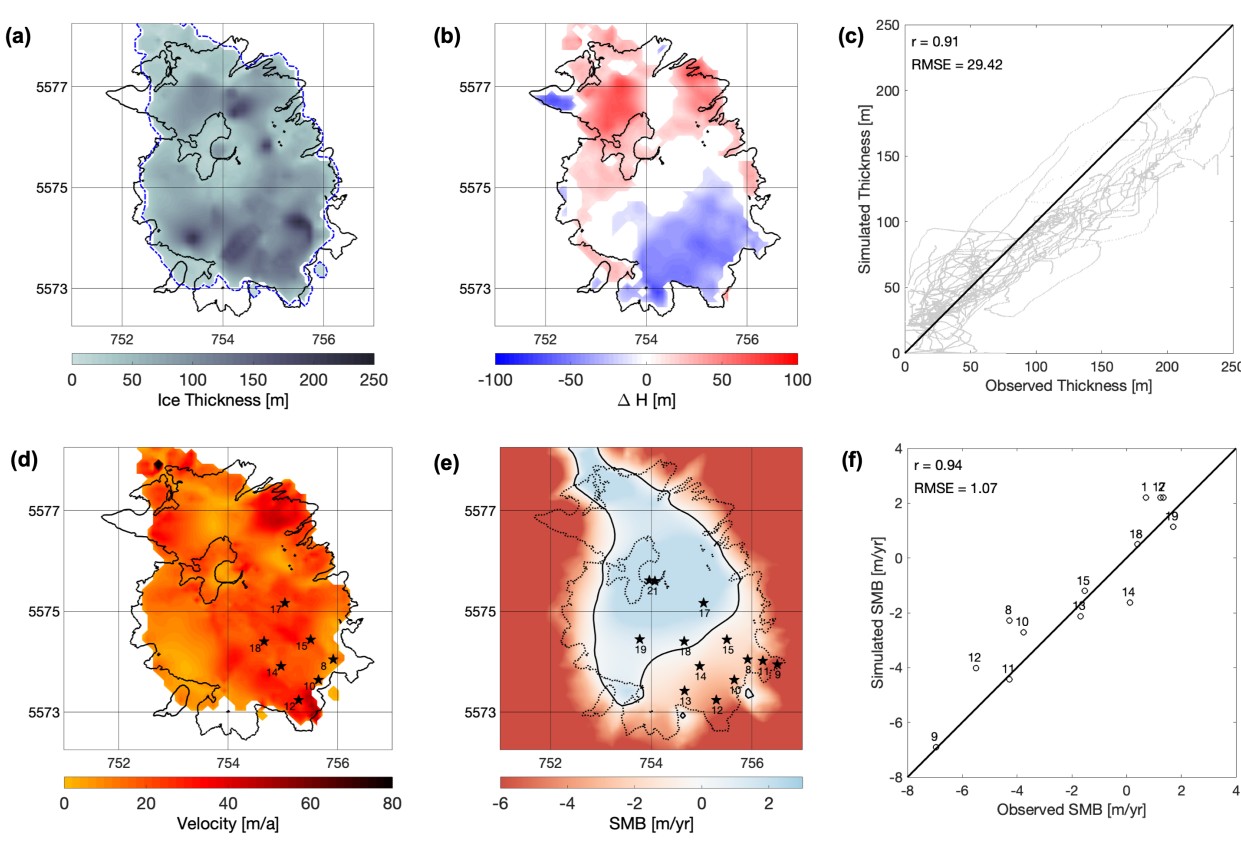

**Figure 6.** Results of the transient spin-up for 2013. (a) Ice thickness distribution with observed (black) and modelled (blue) extent, (b) Difference between modelled and observed ice thickness, (c) Modelled against observed thickness along radar profiles, (d) Surface flow velocity and stakes with velocity observations, (e) Modelled surface mass balance over model domain with SMB stakes as black stars and simulated ELA as solid black line, (f) Modelled against observed SMB at stakes.

### 3.2 Projected future evolution of the ice cap

The evolution of the total ice volume under the RCP2.6, RCP4.5 and RCP8.5 scenarios, as well as the control run with a zero-anomaly with respect to the reference period 2006–2020, is shown in Figure 7. For the control run, the ice cap loses 28% of its volume by 2100, which can be interpreted as the committed loss due to the non-steady-state conditions during the reference period. The projections for the 23 individual climate models (thin lines) can be summarized by the multimodel ensemble mean (thick solid lines) and one-sigma confidence interval (thick dashed lines). All three scenarios start with a negative slope and lose mass at a similar rate, reflecting the present-day negative SMB. From the 2050s, the scenarios begin to diverge significantly, indicating that the differences between temperature increases of each projection start to dominate the ice dynamics. By the end of the century, all mean curves flatten out.

In terms of variability between the climate models, the RCP2.6 scenario starts with a narrow confidence interval which gets larger throughout the century, reflecting disagreements between the ensemble members. For the RCP4.5 scenario, this is only the case until the 2060s, and as the mean curve flattens, the uncertainties remain constant. The uncertainty of the RCP8.5 scenario increases until the 2050s, and then decreases until the year 2100. These contrasts in the projections under different

emission scenarios reflect higher signal-to-noise ratio for the RCP8.5 scenario, as this scenario has a more prominent temperature increase (also see Figure 4). Projected ice volumes and uncertainties for different scenarios and years are summarized in Table 2.

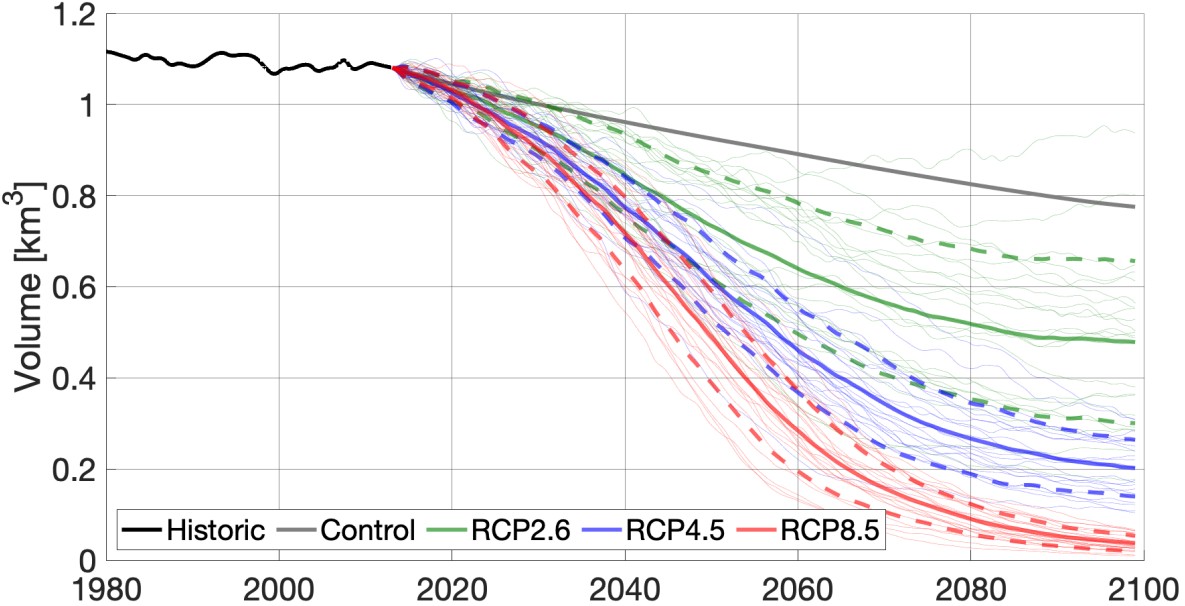

**Figure 7.** Ice volume evolution under the three scenarios RCP2.6 (green), RCP4.5 (blue) and RCP8.5 (red) until the year 2100. Thin lines show the 23 individual evolutions from different climate models, thick solid lines indicate their mean and thick dashed lines the mean plus/minus standard deviation. The solid black line shows the evolution of the transient spin-up between 1979 and 2013, and the thick grey line shows a control run based on a zero-anomaly with respect to the reference period 2006-2020.

| Year | RCP2.6 | RCP4.5 | RCP8.5 |
|------|--------|--------|--------|
| 2013 | 1.08 | 1.08 | 1.08 |
| 2040 | $0.85 \pm 0.09$ | $0.77 \pm 0.07$ | $0.72 \pm 0.08$ |
| 2060 | $0.64 \pm 0.14$ | $0.46 \pm 0.09$ | $0.28 \pm 0.09$ |
| 2080 | $0.52 \pm 0.17$ | $0.27 \pm 0.08$ | $0.09 \pm 0.03$ |
| 2099 | $0.48 \pm 0.18$ | $0.2 \pm 0.06$ | $0.04 \pm 0.02$ |

**Table 2.** Projected ice volumes in $\mathrm{km}^3$ for different scenarios and years: mean and standard deviation obtained by forcing SICOPOLIS for 23 climate models.

In addition to the uncertainty introduced by different climate models, we analyse the impact that the ELA dependence on temperature has on glacier projections. We average the 23 climate model temperature projections for the three scenarios before

running SICOPOLIS instead of forcing it individually with each climate model as in the previous sections. With these mean projections, we perform three model runs for each scenario: the mean gradient between temperature and ELA ($88\,\mathrm{m\,K}^{-1}$), and the upper and lower bound of the one-sigma confidence interval ($51\,\mathrm{m\,K}^{-1}$ and $125\,\mathrm{m\,K}^{-1}$). The resulting ice volume evolutions are shown in Figure 8. The mean curves are very similar to those obtained in Figure 7, however the spread is higher for the RCP4.5 and RCP8.5 scenarios, and lower for the RCP2.6 scenario with respect to those obtained in Figure 7.

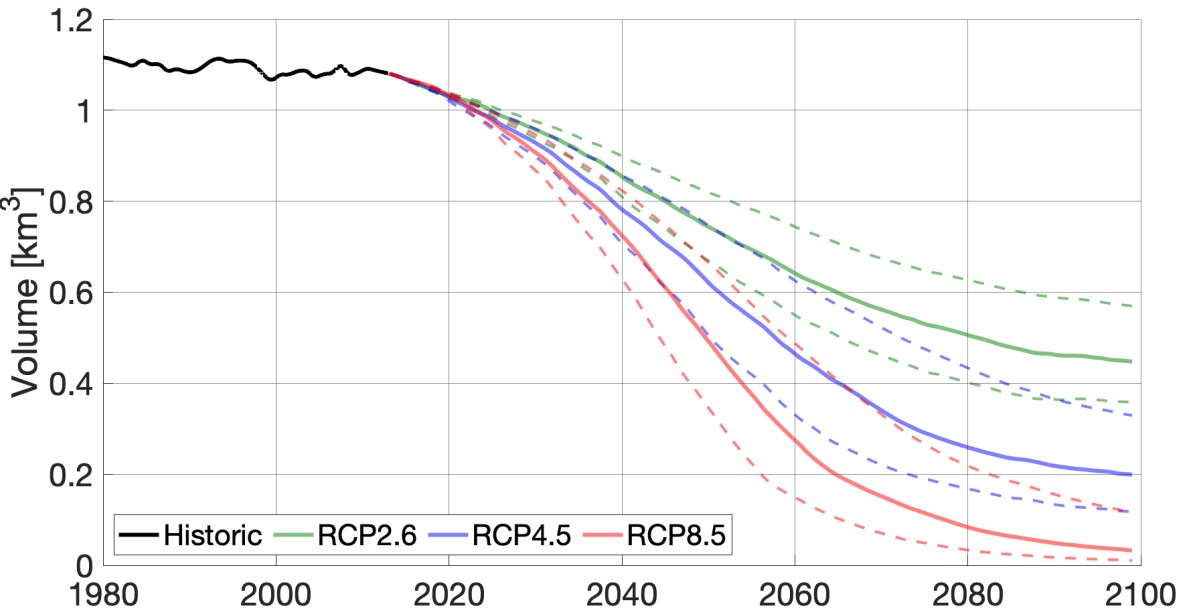

**Figure 8.** Volume evolution for different gradients between ELA and temperature, with the mean shown in solid lines and the one-sigma confidence interval indicated by dashed lines.

The ice volume loss can be broken down into thinning and retreat, i.e. to a lower ice thickness and a smaller ice area, respectively, towards the end of the century. Figure 9 shows the evolution of the ice thickness distribution for the different scenarios, obtained after averaging over all 23 climate models for each of the three scenarios, and Table 3 gives an estimate of thinning by displaying the maximum ice thickness in the same years obtained after averaging over all 23 climate models.

In the RCP2.6 scenario, thinning is dominant until the year 2060, and especially a dramatically reduced maximum ice

thickness is evident until 2040 (see Table 3). After 2060, ice loss becomes less drastic and thinning rates are relatively low until 2100. Retreat is overall moderate and mostly exists in the south-east between 2060 and 2080, presumably as a dynamic response to thinning in the previous decades.

The RCP4.5 scenario shows a stronger retreat pattern compared to the RCP2.6 scenario throughout the century, mostly until 2080. This retreat is accompanied by strong thinning before 2080, and afterwards the reduction in ice thickness is less

pronounced.

The high-end scenario (RCP8.5) shows a clearly different pattern in both thinning and retreat over the 21st century. While the thickness pattern for 2040 is comparable to the two other scenarios, ice loss clearly accelerates between 2040 and 2080. This can be appreciated by the faint colors in the last two plots indicating an ice thickness of mostly under 100 m and a dramatic retreat until the year 2099 (see also Table 3).

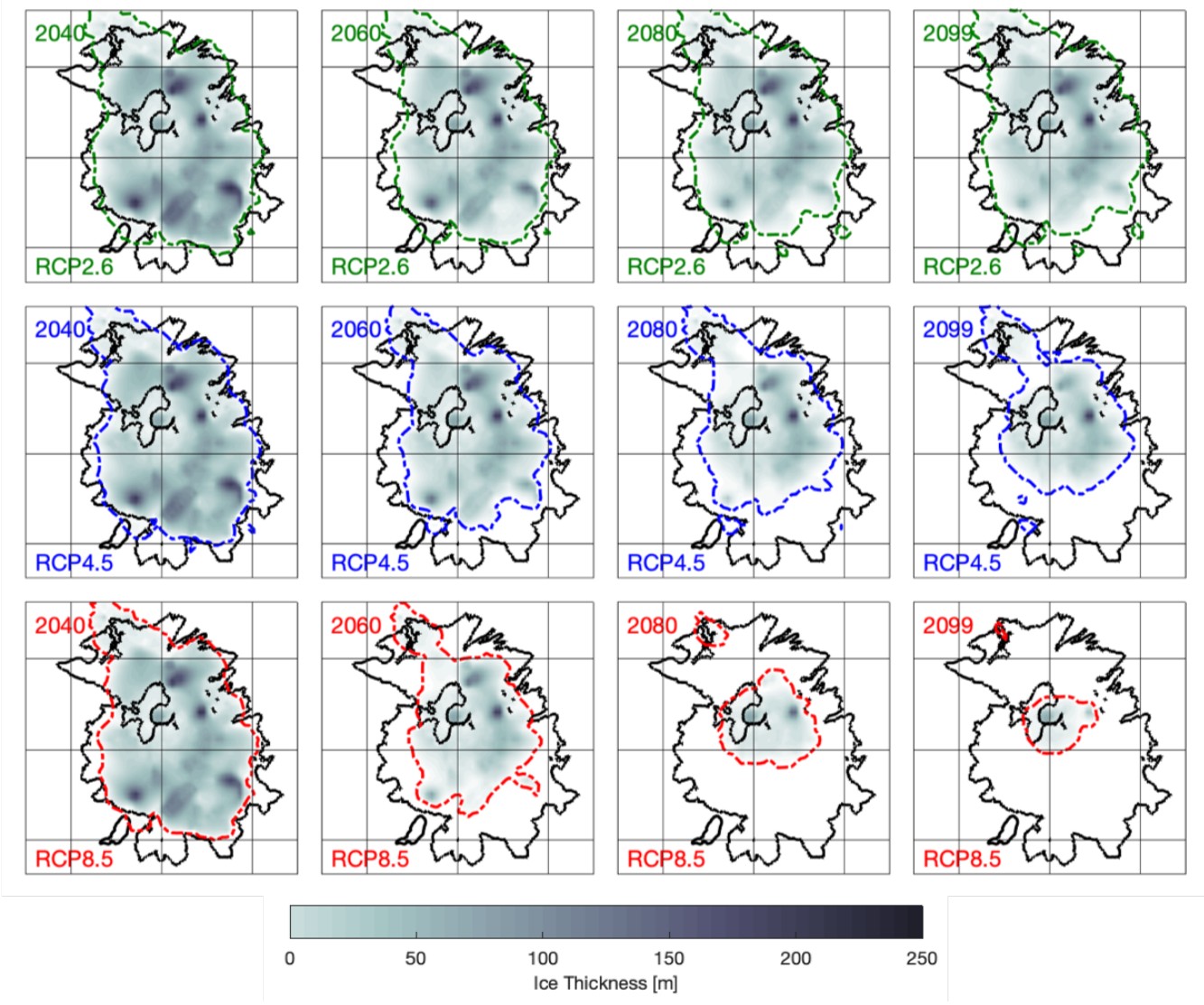

**Figure 9.** Ensemble mean ice thickness for three different future temperature scenarios and four different years, obtained by averaging the thickness obtained from all 23 climate models on every grid cell. The colored dashed lines show modeled ice extent, and black solid lines show the observed ice extent in 2013.

| Year | RCP2.6 | RCP4.5 | RCP8.5 |
|------|--------|--------|--------|
| 2013 | 225.5 | 225.5 | 225.5 |
| 2040 | $191.1 \pm 23.7$ | $187.1 \pm 14.7$ | $186.9 \pm 11.1$ |
| 2060 | $186.9 \pm 45.6$ | $184.7 \pm 45.0$ | $176.9 \pm 24.4$ |
| 2080 | $186.8 \pm 78.1$ | $181.2 \pm 44.0$ | $145.2 \pm 24.8$ |
| 2099 | $186.8 \pm 86.9$ | $178.4 \pm 60.5$ | $91.0 \pm 42.4$ |

**Table 3.** Projected maximum ice thickness in meters for different scenarios and years: mean and standard deviation obtained by forcing SICOPOLIS for 23 climate models.

## 4   Discussion

### 4.1   Present-day simulations

The first part of our study consists of the creation of a present-day steady-state of the Mocho-Choshuenco ice cap. Since drivers of the SMB such as solar radiation and snow redistribution are strongly aspect-dependent, we developed a new SMB parameterization accounting for aspect-dependent SMB variations (see section 2.3). The values of the SMB parameterization were tuned within realistic ranges to find an optimal configuration that reproduces present-day observations of ice thickness, ice extent, SMB, and surface velocity. Here we discuss the physical plausibility of the six tuning parameters. The SMB gradient $M_0 = 0.027\,\mathrm{yr}^{-1}$ was obtained to match the variation of observed SMB at stakes with respect to the elevations, giving a very good match (see Figures 6e and 6f). The direction of maximum ELA ($\varphi_0 = 315°$) was chosen based on the fact that the north-western part is generally more exposed to both solar rdiation and snow erosion by wind, and consequently less melt (more shade) and more accumulation due to snow drift on the south-eastern part. The maximum SMB ($S_0 = 2.2\,\mathrm{m\,a}^{-1}$) was maintained in a range that keeps the stakes at the highest elevations close to the observed values, and fine-tuned in the calibration process. Mean ELA and ELA amplitude ($B_{\mathrm{ELA}} = 2050\,\mathrm{m}$, $A_{\mathrm{ELA}} = 87.5\,\mathrm{m}$) were varied in order to match observations and constrained to maintain a similar ELA in the south-eastern part as observed by Schaefer et al. (2017). This is given with an ELA of $1963\,\mathrm{m}$ in our model, which is near the mean value of $1993\,\mathrm{m}$ obtained from measurements between 2009 and 2013 (Schaefer et al., 2017). As no direct observations of the basal conditions on the ice cap are available, the sliding parameter ($C_b = 1.0 \times 10^{-4}\,\mathrm{m\,a}^{-1}\,\mathrm{Pa}^{-1}$) was purely used as a tuning parameter to match the observations, but its value is within the typical range.

The ice thickness map in Figure 6a reveals that we are able to reproduce the general magnitude of ice thickness (mostly around 100-150 m, with a maximum value of up to 250 m) well. The ice extent is well reproduced, as a comparison between the black and blue lines shows. At the margins, some ice tongues are not recovered, and some others are added. Most notably, this is the case for one of the south-western ice tongues, where stakes B9 and B11 are located. This is a minor inconsistency in our model, and due to our simplified parameterization it would be impossible to recover all details of the observations on the ice cap.

Figure 6b shows the difference in ice thickness between our model and the interpolated ice thickness map in Figure 2a. Ice thickness is mostly underestimated in the south-east and overestimated in the north. However, as Figure 2a shows, there are in fact very few radar measurements especially in the north, and therefore a direct comparison with the interpolation is not meaningful in many places. A more valuable comparison is that in Figure 6c, showing how our model reproduces the directly measured ice thickness along the radar tracks. It shows a satisfying correlation between both with a low RMSE, and most of the simulated thickness values are close to the observed ones. In general, we mostly overestimate the ice thickness in thin areas and underestimate it where ice cover is thick. This might indicate local inaccuracies introduced by our choice of the SIA as a low-order ice flow parameterization, but overall ice thickness is well reproduced.

Modelled ice velocities at the surface are low on the flat parts of the ice cap and get higher towards the outlets of the ice cap (Figure 6d). The simulated velocities at the stake locations are generally lower than the observed ones (Table 1). However, this comparison has to be interpreted with some care as the observed values were taken in October, while the simulated velocities are representative for the whole year. Furthermore, the flow exponent $n = 3$ in Glen's flow law leads to a significant underestimation of surface velocities where thickness is also underestimated. Most stakes lie in areas where the ice is thinner in simulations than that in observations (see Figures 6b and 6d), making this a reasonable explanation. On several stakes (B10 and B15), the velocities are well matched, and we conclude that our spin-up reproduces the observed ice cap well, considering the given observations.

Figure 6e shows the modelled SMB distribution on the ice cap. The only observations available are on the south-eastern catchment, and the distribution of SMB compares well to that of previous observations (see Figure 9a in Schaefer et al. (2017)). Also, simulated and observed SMB at individual stakes match well, as depicted in Figure 6f. Most of the modelled values are very close to the observations, with an RMSE of around $1\,\mathrm{m\,w.e.\,yr}^{-1}$ and a high correlation. As SMB controls the ice evolution in our future projections, these SMB comparisons indicate that our projections are realistic within the observational limitations.

In terms of our choice for a transient spin-up with temperature forcing over the last 35 years, there are several factors that indicate its superiority over a steady-state spin-up where the present-day glacier is built under a constant climate. Most importantly, currently observed SMB is negative (Schaefer et al., 2017), indicating a shrinking ice cap, which by definition could not be reproduced by a steady-state. Our choice of a 35-year transition period is justified by the turnover time of the ice cap, which we calculated as 27 years (see section 2.4). While we do not reproduce the exact glacier state in previous decades, the calculated times indicate that the most important features of the transient state in 2013 should be captured by our model. The control run under a constant 2006-2020 mean temperature in Figure 7 still shows a remarkable shrinking of ice cap in 2100 compared to the most optimistic scenario. This indicates that committed mass loss plays a significant role in future glacier evolution, which could not be represented by a steady-state spin-up. A recent study found highly accelerated glacier mass losses world-wide in the last two decades (Hugonnet et al., 2021), underpinning the need for a transient model initialisation and showing that our projected high mass loss rates in the upcoming decades seem to be more realistic than the more moderate ones that would be obtained after a steady-state model initialisation.

## 4.2 21st century projections

In all scenarios, the future projections start with a significant negative trend, due to the negative present-day SMB. Afterwards, the different scenarios diverge, and in this section we interpret their evolution based on the results presented in section 3.2.

The effect of emission reduction in the RCP2.6 scenario starts to appear around 2050, which correlates well with an estimated response time of 37 years for our ice cap. From the 2050s, ice loss starts to be less drastic for this scenario, and towards the end of the century, the ice cap seems to stabilize at about half its present-day volume. Thinning is more dominant than retreat until 2060, and afterwards retreat takes over, presumably as a dynamic response to the previous thinning.

The uncertainties associated with the volume projections are particularly high for the RCP2.6 scenario, with a large spread
introduced by the different climate models. Therefore, we conclude that it is essential to perform ice cap projections with an ensemble of climate models rather than a single model, in order to avoid bias towards the underlying assumptions of one particular model.

The RCP4.5 scenario assumes a significant reduction of emissions only after the 2040s, and this reflects in our results by the fact that ice volume steadily decays until the around 2080, and only then becomes more stable. Apart from the reduced
emissions, another explanation for the flattening of the curve is the fact that by 2080 most of the plateau of the ice cap has melted away, and further elevations of ELA have less influence due to the steep slopes around the summits. This interpretation is confirmed by the ensemble uncertainty, indicating a generally good agreement between the climate models with regards to the state of the ice cap at the end of the century. As opposed to the RCP2.6 scenario, retreat sets in earlier and accompanies the thinning that is prevalent during the whole 21st century.

In the RCP8.5 scenario, assuming no emission reduction at all, the ice volume loss becomes much steeper from the 2030s, losing quickly most of the mass of the ice cap. This mass loss is driven by both high retreat and thinning rates. Only after 2080, with around 10% of the initial ice volume left, losses start to become less when the ELA retreats towards the summit. By the year 2100, the only remaining patches of ice are very close to the Mocho summit. The ensemble uncertainty for this scenario is highest during the extreme volume loss in the middle of the century, and becomes very small towards the end of the century,
indicating that most climate models agree on the almost complete disappearance of the ice cap.

## 4.3 Limitations of our approach

In this study, the principal uncertainties we assign to our results are based on the spread of the temperature projections of the global climate models and on the uncertainty of the temperature-ELA parameterization. In this section, we discuss possible further sources of uncertainty, and make suggestions on how future work could encounter these challenges.

Our approach is based on the shallow ice approximation (SIA), with assumptions including almost parallel and horizontal glacier bed and surface, significantly larger horizontal than vertical dimensions and simple-shear ice deformation. While these assumptions hold well for the large Greenlandic and Antarctic ice sheets, it is less obvious that the SIA can be employed on such a small study object as the Mocho-Choshuenco ice cap. The SIA assumptions are violated especially in the steep regions around the two summits and towards the boundaries of the present-day ice cap. However, they hold true for large parts of the

plateau which is the most important area in our future projections. Previous studies have suggested that low-order assumptions such as the SIA hold well for glaciers whose behaviours are mostly driven by SMB (Adhikari and Marshall, 2013), which is the case for the Mocho-Choshuenco ice cap. However, it would be a valuable experiment to reproduce our results with a full-Stokes model such as Elmer/Ice to verify the applicability of the SIA.

Knowledge about the bed of the ice cap is essential to perform ice flow simulations. We created a bed map based on present-
355  day topography and a number of ground-penetrating radar profiles published by Geoestudios (2014). Even though these profiles cover a significant portion of the ice cap, there are large gaps in data coverage, especially in the north-western part of the ice cap. More observations could help to reduce the uncertainty introduced by these gaps.

Regarding the ELA gradient we use to relate temperature increase to glacier SMB, it is important to note that we have only few data points given for this relationship (Schaefer et al., 2017). With more years of ELA-temperature pairs and a thorough
uncertainty estimation, we could achieve a higher confidence in our ELA gradient. However, by performing the simulations for the mean gradient and a lower and upper bound, we are within the range of most previous studies (e.g. Six and Vincent, 2014; Sagredo et al., 2014; Wang et al., 2019).

Another significant limitation lies in the SMB parameterization. While the new aspect-dependent parameterization was able to improve the reproduction of the present-day ice cap significantly, there is still space for improvement. Especially the
northern part is still not well reproduced by SICOPOLIS, and it might be advantageous to extend the new parameterization to the Choshuenco peak. In order to verify our parameterization, it would be helpful to obtain SMB measurements in the north-west, i.e. between both summits, and thus extend the stake network that is currently focused on the main catchment in the south-east of Mocho summit. This could provide more observational constraints on the ELA difference between the north-west and south-east.

Another way of producing more realistic SMB maps for the ice cap would be using explicit models that try to quantify the physical processes which determine glacier mass balance, e.g. the COSIPY Model (Sauter et al., 2020). A drawback of these complex models is that they need many input parameters (such as precipitation, relative humidity or wind speed) with a high spatial resolution. These can be obtained by regional climate model simulations (e.g. Bozkurt et al., 2019). However considerable uncertainties are associated to these simulations and a careful validation of the results is necessary before using
them as drivers of SMB simulations. Additionally, only few high resolution regional climate simulations are available in the moment which is why we prefer our simple temperature dependent SMB parameterization combined with a multi-model approach using 23 different GCMs as drivers of our simulations.

### 4.4 Global context of glacier decline

To our knowledge, there are only few previous studies that have projected the future evolution of glaciers in the Andes. The
nearest study object to the Mocho-Choshuenco ice cap is the Northern Patagonian Icefield for which until 2100 an ice mass loss of 592 Gt has been projected under the A1B scenario which is comparable to the RCP6.0 scenario, and therefore between our results for RCP4.5 and RCP8.5 (Schaefer et al., 2013). Relating this ice loss to more recent estimates of total ice mass (Carrivick et al., 2016; Millan et al., 2019), around 50% of the ice mass are projected to disappear. However, these simulations

were performed on a fixed geometry, and therefore considered only changes in SMB, making it difficult to compare their results to ours. Collao-Barrios et al. (2018) obtained a committed mass loss of approximately 10% for San Rafael Glacier under current climate, significantly less than the 28% which we estimated. However, they maintained a constant glacier area during their simulations and therefore neglect glacier retreat, which could dramatically change rates of frontal ablation.

Möller and Schneider (2010) projected the future evolution of Glaciar Noroeste, an outlet glacier of the Gran Campo Nevado ice cap in southern Patagonia between 1984 and 2100. Their projections were made for the B1 scenario and yielded a volume loss of around 45% which is significantly less than the 61% volume loss that we project for the comparable RCP4.5 scenario between 2013 and 2100. Their results are based on a calibrated relationship between area and volume, and not on ice flow modelling as our study.

Hock et al. (2019) and Marzeion et al. (2020) are two studies who have projected 21st century glacier evolution world-wide using six and eleven different different glacier models, respectively. In both studies, the southern Andes are one of the study areas, and they both predict rather low mass losses of around 20% for the RCP2.6 scenario and under 50% for the RCP8.5 scenario, which is considerably less than ours (55% for RCP2.6, 97% for RCP8.5). However, making a direct comparison between these studies and our results is problematic for several reasons. First, their study region is highly dominated by the large Patagonian icefields, where many glaciers terminate in ocean or lakes, with frontal ablation contributing to 34% of overall mass loss (Minowa et al., 2021). Frontal ablation, however, is only parameterized in one of six (Hock et al., 2019) and two of eleven models (Marzeion et al., 2020), and their results need therefore to be interpreted with care. Second, SMB in these global models is highly simplified and averaged over a huge amount of glaciers. While this is convenient in obtaining satisfactory global projections, the accuracy is likely limited on a regional or local scale. In fact, SMB is positive on the Southern Patagonian Icefield (Schaefer et al., 2015), reinforcing the need to account for frontal ablation when estimating mass losses. In the case of our small ice cap, many detailed SMB observations are available, and our results therefore yields valuable local-scale estimates of SMB and future mass loss against which global models such as those in Hock et al. (2019) and (Marzeion et al., 2020) can be calibrated.

The only glacier in the Andes for which future projections under climate change scenarios are available, based on simulations with an ice-flow model (Elmer/Ice), is Zongo Glacier in Bolivia Réveillet et al. (2015). They projected 40% and 89% volume loss for the RCP2.6 and RCP8.5 scenarios, respectively. The value for the high-end scenario is comparable to ours (97%), which might be expected as both glaciers are about to disappear by the end of the century, and therefore have already lost the majority of their ice mass relative to their present state. Our projections for RCP2.6 ($55 \pm 16\%$) are also within the range of their RCP2.6 projections. However, this comparison needs to be treated with care due to the climate differences between the tropics and the Wet Andes, and also due to the higher ELA gradient with temperature of $150\,\mathrm{m\,K^{-1}}$ used in their study in comparison to $88\,\mathrm{m\,K^{-1}}$ used in our study. Another factor that changes from glacier to glacier are the geometric conditions, which can have significant impact on volume losses.

Outside the Andes, only few studies have projected glacier evolution in the 21st century with ice flow models. Among them is that of Adhikari and Marshall (2013) who performed ice flow simulations on Haig Glacier in the Rocky Mountains and projected the disappearance of the glacier by 2080 under the RCP4.5 and RCP8.5 scenarios. In Europe, Jouvet et al. (2011)

projected a volume loss of 90% for Grosser Aletschgletscher in Switzerland until 2100 under the A1B scenario, and indicated
that even under the present climate the glacier is in disequilibrium and would continue to lose significant amounts of ice. Wang
et al. (2019) investigated the future evolution of Austre Lovénbreen with the full-Stokes ice flow model Elmer/Ice, a mountain
glacier in Svalbard, and found that with an intermediate temperature increase scenario the glacier would disappear by 2120,
and by 2093 for the most pessimistic scenario.

Even though different model setups and parameterizations were applied for all glaciers in the mentioned studies, most
of them show a similar trajectory for the glacier evolution in the next 60 to 100 years, and our projections for the Mocho-
Choshuenco ice cap fit well into them. All of them lose a high percentage of ice mass during the 21st century and we can expect
many mountain glaciers in different parts of the world to disappear in the first half of the 22nd century, without reductions of
greenhouse gases.

## 5   Conclusions and outlook

In this study, we applied the ice sheet model SICOPOLIS to reproduce the current state of the Mocho-Choshuenco ice cap
and to project its future evolution under different emission scenarios. To our knowledge, this is the first estimate of future
glacier evolution obtained from an ice flow model forced with climate change scenarios for the Wet Andes, and the second for
the whole Andes. Using a linear temperature-ELA parameterization, we investigate the future of the ice cap using projected
temperature changes from 23 GCMs as input. A considerable spread of the projected ice volume at the end of the 21st century
is obtained, depending on the emission scenario and GCM.

The mean projected ice volume losses until the end of the century are $56 \pm 16\%$ (RCP2.6), $81 \pm 6\%$ (RCP4.5) and $97 \pm 2\%$
(RCP8.5) with respect to the ice volume derived from measurements in 2013. This means that even under the most optimistic
emission scenario the expected loss of ice volume is between 40% and 72%. The spread between the results, when driving the
model by different GCMs, becomes lower when considering higher emission scenarios: under the emission scenario RCP8.5,
which does not consider a reduction of our emission of greenhouse gases, it is likely that the ice cap will lose more than 95% of
its current volume by 2100. Since temperature projections are relatively uniform in the region and geometry of the surrounding
ice caps are similar to Mocho-Choshuenco ice cap, we can expect similar projections of high volume losses for other ice caps
in the Chilean Lake District (39-41.5°S).

The Mocho-Choshuenco ice cap is the smallest ice body to which SICOPOLIS has been applied so far, justified a priori by
the cap-like geometry (as opposed to, for example, valley glaciers), and a posteriori by the reasonably good performance of
the model in replicating the present-day ice cap. Nevertheless, it would be valuable to check if the application of a full-Stokes
glacier flow model (as for example Elmer/Ice, Gagliardini et al. (2013)) affected the simulated state of the ice cap notably, or
if the disagreements are mainly caused by our simplified SMB parameterization.

When trying to project the future of the largest ice bodies of the Southern Andes (the Patagonian icefields), the interaction
of their outlet glaciers with the surrounding water bodies becomes crucial. Adequate parameterizations for frontal ablation are

necessary, which allow the glaciers to adapt their frontal positions according to the glacier flow, which, in turn, will be crucially determined by its interaction with the water bodies.

*Code and data availability.* SICOPOLIS is free and open-source software, available through a persistent Git repository hosted by the Alfred Wegener Institute for Polar and Marine Research (AWI) in Bremerhaven, Germany (https://gitlab.awi.de/sicopolis/sicopolis). Detailed
instructions for obtaining and compiling the code are at http://www.sicopolis.net. The output data produced for this study will be made available at Zenodo, https://doi.org/10.5281/zenodo.5053396

.

## Appendix A: Global climate models

Table A1 gives details on the 23 global climate models that were used to force SICOPOLIS with future temperature projections.

| IPCC Model ID | Institution | Resolution (degree) (LonxLat) |
|---|---|---|
| BCC_CSM1_1 | Beijing Climate Center, China Meteorological Administration, China | 2.81x2.77 |
| BCC_CSM1_1_M | | 1.12x1.12 |
| BNU_ESM | College of Global Change and Earth System Science, Beijing Normal University, China | 2.8x2.8 |
| CanESM2 | Canadian Centre for Climate Modelling and Analysis, Canada | 2.81x2.79 |
| CCSM4 | National Center of Atmospheric Research, USA | 1.25x0.94 |
| CESM1-CAM5 | | 1.25x0.94 |
| CNRM_CM5 | National Center of Meteorological Research, France | 1.41x1.40 |
| CSIRO_Mk3_6_0 | Commonwealth Scientific and Industrial Research Organization (CSIRO), Australia | 1.875x1.86 |
| FIO_ESM | The First Institute of Oceanography, SOA, China | 2.8x2.8 |
| GFDL_CM3 | | 2.5x2.0 |
| GFDL_ESM2G | NAOO Geophysical Fluid Dynamics Laboratory, USA | 2.5x2.0 |
| GFDL_ESM2M | | 2.5x2.0 |
| GISS-E2-H | Goddard Institute for Space Studies, USA | 2.5x2.0 |
| GISS-E2-R | | 2.5x2.0 |
| HadGEM2-AO | Met Office Hadley Centre, UK | 1.875x1.25 |
| IPSL-CM5A-MR | Institut Pierre-Simon Laplace, France | 2.5x1.25 |
| MIROC5 | Atmosphere and Ocean Research Institute (The University of Tokyo), National Institute for Environmental Studies and Japan Agency for Marine-Earth Science and Technology, Japan | 1.41x1.39 |
| MIROC_ESM | | 2.81x1.77 |
| MIROC_ESM_CHEM | | 2.81x1.77 |
| MPI_ESM_LR | Max Planck Institute for Meteorology, Germany | 1.875x1.85 |
| MPI_ESM_MR | | 1.875x1.85 |
| MRI_CGCM3 | Meteorological Research Institute, Japan | 1.125x1.125 |
| NorESM1_M | Norwegian Climate Center, Norway | 2.5x1.875 |

**Table A1.** Details of global climate models used in this study. For further information on CMIP5 and the individual models, see Taylor et al. (2012) and references therein.

*Author contributions.* Marius Schaefer, Ralf Greve, and Matthias Scheiter designed the study. Ralf Greve developed the ice-sheet model SICOPOLIS. Matthias Scheiter ran the simulations with advice from Marius Schaefer and Ralf Greve. Eduardo Flández contributed to the simulations. Deniz Bozkurt provided the temperature projection data and assisted with the ice cap projections. Matthias Scheiter wrote the manuscript with contributions from all authors. All authors discussed and interpreted the results.

*Competing interests.* The authors declare that no competing interests are present.

*Acknowledgements.* We are grateful to Gino Casassa for providing the ice thickness data used in this study. Discussions with Andrew Valentine, Buse Turunçtur, and Shubham Agrawal helped to improve this paper. We thank Klaus Spitzer and Malcolm Sambridge for their generous support in undertaking this research. We acknowledge the World Climate Research Programme Working Group on Coupled Modelling, which is responsible for CMIP, and we thank the climate modeling groups (listed in Table A1) for producing and making available their model output. We thank the editor Benjamin Smith for handling the manuscript and two anonymous referees whose comments helped to improve the quality of the manuscript.

Matthias Scheiter acknowledges financial support from the Australian National University and the CSIRO Deep Earth Imaging Future Science Platform. Marius Schaefer is supported by the FONDECYT Regular Grant, Etapa 2018, grant no. 1180785. Eduardo Flández acknowledges support from FONDECYT grant no. 1201967 and a doctoral fellowship from ANID. Deniz Bozkurt acknowledges support from CONICYT-PAI 77190080, ANID-PIA-Anillo INACH ACT192057 and ANID-FONDECYT-11200101. Ralf Greve was supported by Japan Society for the Promotion of Science (JSPS) KAKENHI grant Nos. JP16H02224, JP17H06104 and JP17H06323, by a Leadership Research Grant of Hokkaido University's Institute of Low Temperature Science (ILTS), and by the Arctic Challenge for Sustainability projects ArCS and ArCS II of the Japanese Ministry of Education, Culture, Sports, Science and Technology (MEXT) (program grant numbers JPMXD1300000000, JPMXD1420318865).

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
