# Peer review of "21st century fate of the Mocho-Choshuenco ice cap in southern Chile"

_The Cryosphere, 2020_

## Referee Comment (RC1) · Anonymous Referee #1 · 12 Jan 2021

This study uses an ice-flow model, glaciological observations, and climate-model output to project the evolution of Mocho-Coshuenco ice cap in the 21st century. The stated goals are to calibrate the model such that it reasonably captures the current state, and use it to project future ice loss under different forcing scenarios. In their projections, the authors find a range of outcomes depending on forcing scenario, namely relatively little ice loss under the RCP 2.6 scenario, and near total loss of the ice cap by 2100 under RCP 8.5.

As the authors note, there are relatively few glaciological modeling studies targeting this area, and even fewer have used models that explicitly incorporate ice dynamics, so a case study on this glacier complex has the potential to add useful understanding of glacier change in the region. This setting is a good target for applying and evaluating

a 2D ice-flow model (SICOPOLIS), as there exist ice thickness, mass balance, and velocity data to aid model calibration. I think the choice of model is appropriate for the setting and think the overall framework of using these observations and model together is promising and worthy of investigation.

However, in my view, some fundamental considerations for calibrating and initializing the model are missing and/or flawed, giving me concerns about whether it is indeed calibrated well enough to yield reliable projections into the 21st century. I detail these below, but briefly, they are (1) whether the mass-balance parameterization is consistent with observations; and (2) the assumption of a steady-state to initialize the model in the early 21st century, when these glaciers have already been responding to industrial-era warming. Both of these factors could affect the dynamical response of the simulated ice cap.

Given the stated goals of the paper, I think these factors need to be evaluated before this study can be published. I am not sure whether this will call for re-running the projections with a re-calibrated model, but as it is described now, I am not yet confident in the projections beyond the qualitative conclusions (e.g., massive loss of ice under RCP 8.5 and a significant contrast between RCP scenarios).

I expand on these issues below, and also provide minor and technical comments. I hope the authors will consider these factors, as I do think they have the tools set up for a nice modeling study and a positive contribution – but I think the methodology does need to be adjusted.

Major comments

1) Calibration procedure and surface-mass-balance (SMB) scheme

My first concern is that the parameter calibration isn't adequately evaluated against available mass balance observations. I would argue that matching the overall volume and/or extent is not necessarily a sufficient constraint to conclude the model captures the most important features of the present-day glacier (as is concluded at several points). Given that there are mass balance observations available (already used to calibrate the sensitivity to future warming), there is yet more information that could be used to evaluate the calibrated model parameters and the initial glacier state that they yield.

In particular, the calibrated value for maximum snowfall (S0 = 1.07 m/yr) strikes me as surprisingly low given the high precipitation and mass turnover rates discussed earlier in the paper. The mass balance gradient (line 167) is fixed at a (quite high) value of M0 = 0.023 yrˆ-1 (2.3 m w.e./yr per 100m vertical?). How was this chosen?

One thing that strikes me from these values of S0 and M0 is that mass balance must plateau very quickly (< 50 vertical meters) above the ELA, implying a large area where mass balance is uniform at the maximum value. Is this backed up by the available mass balance data? From my own look at Schaefer et al., (2017), it appears that both accumulation and ablation are substantial all the way to the summit in the seasonal balances (e.g., their Fig. 7). If there is in fact non-zero ablation over the entire glacier surface, the entire surface should be susceptible to surface-mass-balance anomalies caused by further melt-season warming. However, if I understand the existing SMB scheme correctly, the model assumes warming doesn't actually cause a SMB anomaly in grid cells above the maximum snowfall cutoff. And, as noted above, this seems like a large initial area, due to the values of S0 and M0. Should warming really only affect the lower reaches of the glacier?

Obviously, no simulation can be expected to capture every detail of the mass balance, and I think a simple elevation/aspect dependent scheme can be a reasonable approach. But as described it seems like there are some embedded assumptions that may not be consistent with observations. The pattern of mass balance and the pattern of anomalies driven by warming should be include in the evaluation of optimal model parameters, given that stake data are available. If I have misinterpreted the SMB scheme, please clarify.

A figure showing the initial spatial pattern of SMB could really aid the reader in interpreting how the maximum snowfall and aspect-dependent ELA affect SMB on the actual topography (e.g., beyond the schematic in Fig. 3).

2) Assumption of steady-state for spinup

My second major comment is on the steady-state spinup for the initial condition. Assuming a steady state is questionable, given more than a century of global warming over the industrial era, to which glaciers respond with a dynamical lag. This lag means that glaciers, in general, are out of equilibrium with current climate (see, e.g., Lüthi et al., 2010; Christian et al., 2018; Marzeion et al., 2018; many others). Forcing a steady state can throw off parameters in the initial calibration (e.g., the ELA), and could throw off the initial transient response when forcing is applied.

The observations of negative mass balance (noted on line 58; from Rivera et al., 2005 and Shaefer et al., 2017) are themselves an indication of disequilibrium, and another reason to include SMB in the model calibration (granted, 5 years is not many observations to define a mean balance). The Rivera et al. study also shows substantial retreat since at least 1976, which would also suggest that the early 2000s extent is not likely to be a steady state.

At the very least, I think it is necessary to estimate how far from steady state the ice cap is at the time the simulation starts, in case the projections need to be qualified in light of this assumption, or the model recalibrated. Some first-order estimates could be made based on the ice cap's estimated response time (e.g., $H/b\_terminus$, see Johannesson et al., 1989). If the response time is long and disequilibrium substantial, it would be necessary to start the simulation earlier to properly capture the transient response for future projections. This is especially true for evaluating the difference between RCP 2.6 and 8.5 trajectories, as the "committed" response to past warming may be a substantial part of the true 2.6 trajectory (with little additional warming), but this would not be captured if the model starts in a steady state in the 2000s.

Minor comments (line by line)

- Line 23: change 10000 mm/yr to 10 m/yr ? (That is a lot of zeros for the readers' eye!)

- 27: "As these are best represented in ice-flow models, they are..." Clarify wording: "these" and "they" presumably refer to different things here, but sentence is ambiguous

- 59-61: how are you defining mass turnover here? Can you elaborate on how these temperature measurements indicate this?

- 87: "we replaced solving the energy balance by.." somewhat awkward wording, consider rephrasing

- 100: "SMB should be lower" ... is it? Based on observations or simulations? It would be helpful to discuss what processes likely lead to this pattern, to help the reader understand how much the model may be simplifying reality.

- 100–104 and Fig. 3b: I'd suggest making the angle in the schematic correspond to the angle used in the actual simulations. I initially got confused as the wording in the text (referring to Mocho) doesn't correspond to the orientation in the schematic.

- 107: if the ELA is defined by the angle with respect to the summit, and the mass balance gradient is constant, doesn't this lead to very sharp spatial variations in mass balance as points near the summit? I suppose the maximum snowfall could limit this, but is this a realistic pattern? Again, a spatial map of SMB could be useful for the reader.

- 119: Clarify: model mean, time mean or both?

- 136: Is precipitation taken into account for this regression? That will affect ELA variability too... It is one thing to only consider temperature for the future projections (but see later comment), but I would think the effect of each year's accumulation should be taken into account to calibrate this relationship, especially with only 4 years available.

- 136: Also, is there a particular reason the temp-ELA relationship and future projections are based on annual-mean rather than melt-season temperatures? Do the climate models predict melt-season temps warm at the same rate as annual mean?

- 167: Again, where are the values for M0 and phi taken from?

- 172: "does not reproduce well" ...Consider rewording for clarity.

- 177: 200 m/yr is very fast! Can you comment on why the model might give velocities an order of magnitude higher here than most values in Table 1? Is it the geometry, or SMB pattern that allow this?

- 181: I'm just curious if you know why there are seasonal but not annual velocities? I'm surprised annuals weren't derived from, e.g., mass balance stake locations. Were seasonal velocities only measured in one year?

- 183: As I understand from the SMB parameterization, precipitation is implicitly assumed to not change. Is this roughly consistent with the model projections and/or observed trends for the area? I'd expect that temperature is the main forcing, but recommend at least stating that this assumption is made.

- 187: Retreat is strongest in the north for RCP 2.6... is this partly because of the imposed higher ELA and cap on mass balance?

- 215–17: I find this statement on internal variability confusing. Do you just mean that the spread due to different climate models and scenarios hasn't had time to diverge? Consider rewording for clarity.

- 220: When considering a different ELA-temperature relationship, doesn't this imply other parameters are also different (e.g. the vertical SMB gradient?). Does the initial state reflect differences in these parameters, if any?

- 238: "high observed velocities" ... do you mean high modeled velocities (referring to an area without extant ice)

- 248: I think you mean thinner here, right?

- 246—48: Here you have explained the low velocities in terms of anomalously thin ice, but why is the ice too thin? The combination of too-thin and too-slow together indicate that overall fluxes are underestimated in these areas. . . which ultimately seems like a mass balance issue. Could this be related to the rather low cap on mass balance (see major comment above)?

- 253—54: I find it a bit circular to invoke a "stable position at the moment" to explain a model result, when the stable position is imposed by your choice of a steady-state initial condition. This is one area where the steady-state assumption (see major comment) can affect projections.

- 266: Suggest word choice other than "unstable". . . there's no instability in a dynamical sense here, just a larger forcing.

- 316: "would have disappeared" » projected to disappear?

- 319: "maintained glacier area constant" » "maintained a constant glacier area" ?

- 330: "lost majority of their ice mass" . . . relative to preindustrial?

- 331: "The" » this

- 331-332: general comment here that different proportions of volume lost over a given timeframe can be due simply to different glacier geometries/hypsometries. I think that should be borne in mind for all of the comparisons in this section. . .

- 357: missing period

- 370: then » than

References (beyond those already cited in manuscript)

Christian, J. E., Koutnik, M., & Roe, G. (2018). Committed retreat: controls on glacier disequilibrium in a warming climate. Journal of Glaciology, 64(246), 675-688.

Jóhannesson, T., Raymond, C., & Waddington, E. D. (1989). Time–scale for adjust-

ment of glaciers to changes in mass balance. Journal of Glaciology, 35(121), 355-369.

Lüthi, M. P., Bauder, A., & Funk, M. (2010). Volume change reconstruction of Swiss glaciers from length change data. Journal of Geophysical Research: Earth Surface, 115(F4).

Marzeion, Ben, et al. "Limited influence of climate change mitigation on short-term glacier mass loss." Nature Climate Change 8.4 (2018): 305-308.

---

## Referee Comment (RC2) · Anonymous Referee #2 · 24 Jan 2021

Scheiter and co-authors present a model for the mass balance and ice flow of the the Mocho-Choshuenco ice cap, and apply it to produce projections of mass change during the 21st century. Projections of glacier mass change are generally relevant, as glaciers contribute strongly to sea-level rise, and to seasonal water availability in some regions of the world. In the paper, it remains a bit unclear what is the motivation to study this specific glacier, given that a number of models (not all of them of much lower complexity) have projected glacier mass change globally. Either the specific interest of this particular glacier, or the specific advantages of the methods used here over those used in the other projections of the glacier need to be clearly stated and evaluated by the authors to demonstrate the interest of the study to a wider audience.

Additionally, there are two major concerns that call the results into question (both of

them detailed below): (i) the assumption of steady-state used by the authors for the spin-up of the glacier model, which is in stark contrast to the observed mass change rates; (ii) the complete lack of evaluation of the model's mass balance results. Because of the combination of these two issues, I would be very surprised if the results are not flawed by a very substantial positive mass balance bias, such that the mass loss projections are substantially too low.

Addressing these issues would be possible (see below for a few specific suggestions), but it would require not only new experiments, but also a completely new rationale of model setup and evaluation. This exceeds the scale of work that I would consider a "major" revision, such that I would recommend to address this in a new submission, if the authors decide to attempt it.

General/major comments:

- Assumption of steady-state for spin-up: the paper starts by introducing the glaciers of the southern Andes having among the highest mass losses of all glacier regions worldwide, and specifies a SMB of almost -1 m w.e./year in observations for the Mocho-Choshuenco ice cap (L58). However, the method assumes a zero mass balance of ice cap under present-day (2006-2020) conditions, by requiring that the ice cap's geometry is closely reproduced by the model as a steady state at a temperature anomaly of zero. This is a strong internal inconsistency which is currently not at all addressed in the paper. Closely related is the lack of discussion of the parameter values obtained by matching the steady-state thickness as closely as possible to observations: equation 9 indicates B_ELA = 1777 m from the observation as opposed to 2035 m from the observation, which (again according to eq. 9) corresponds to a temperature offset of almost 3 K ((2035 m-1777 m)/88m/K). The turnover of 5 m w.e./year (L59) is in apparent contradiction to a maximum annual snow fall S_0 of about 1 m as best parameter values (L166). The initialization of an ice flow model is a complex task, but has been addressed before (e.g., Eis et al. 2019, DOI: 10.5194/tc-13-3317-2019; Zekollari et al. 2019, DOI: 10.5194/tc-13-1125-2019). These studies may be helpful for coming up

with an adequate initialization approach.

- Lack of validation of modeled SMB: the authors chose the somewhat unusual way to calibrate parameters of the mass balance equation through matching observed and modeled ice thickness, which I would assume are closer related to parameters of the ice flow model (which are also included in the observation). I don't understand the rationale of this approach, given that mass balance observations are available, and could easily be used for optimizing the mass balance parameters. At the same time, an evaluation of the model's performance concerning SMB is completely lacking. Since a steady state condition is used for the recent past as spin-up, and observations of the SMB during a very similar period are 0.9 m w.e./yr, I suspect that the model has a positive bias of around +0.9 m w.e./yr (not exactly, because the observations only cover a fraction of the glacier's surface). If these presumptions are correct, this would imply that also the projections have a strong positive SMB bias, such that they would strongly underestimate the future rate of mass loss. It is good that the authors evaluate their results against ice thickness and velocity observations, but with the application in projections of mass change in mind, the evaluation of the SMB results is even more important. Without a convincing evaluation the projections cannot be trusted.

- Discussion of comparable studies: At two occasions in the manuscript (L29ff L312ff, the authors state that there are few studies that have projected the future evolution of glaciers in den Andes). Among the studies they cite is Hock et al. (2019), which alone summarizes six studies; a more recent intercomparison is Marzeion et al. (2020, DOI: 10.1029/2019EF001470), which includes seven different models. These are additional to the ones discussed in the paper, but very different in that they don't focus on one (or a few) individual glacier(s), but include all glaciers worldwide. I think it is possible to turn this study into a publishable paper even though by now, there are many models around that have been applied to this specific glacier. But it will be necessary to go into the individual model publications (not just the intercomparison paper, as done now) and see how they are approaching the problem, and discuss the merits of the approach
used here approach in this context: what are the advantages of their mass balance parameterization over those used in the models summarized in Hock et al. (2019) and Marzeion et al. (2020)? What are the advantages of using SICOPOLIS instead of the (mostly simpler) approaches in the global models?

I am convinced that once the first two major issues are addressed by the authors, the results will change substantially. I have therefore abstained from providing detailed/minor comments to the sections that present or discuss these results. These should be addressed at a later stage, if the authors decide to revise and resubmit the paper.

Specific/minor comments/suggestions

- L14: I don't see this generalization backed up by the study results.

- L24: I assume there is a strong seasonality in this number; it would be helpful to be a bit more specific.

- L54: repetitive, can be shortened.

- L61: it can be explained by the climatology, as documented in the data – not the data itself.

- L62: based on the setting of the station and the glacier (orography, wind direction, etc.) would you expect precipitation at the glacier to be higher or lower than in Puerto Fuy? A qualitative assessment would be helpful for readers unfamiliar with the area.

- L76: is an uncertainty assessment available either the total volume?

- Fig. 3b and discussion around it is a bit confusing. Assuming that $A\_ELA$ and $B\_ELA$ are both positive, and that y is latitude, the maximum ELA would be in the north-east sector. However, the text says the SMB should be lower (equivalent to a higher ELA) in the north-western sector. Since you prescribe $phi\_0$ anyway (L167), why not make Fig. 3b using the actual values used?

- L136: by the ELA, $M\_0$ and $S\_0$.

- Eq. 5 and following: I'm unfamiliar with this notation. Please explicitly define N. Shouldn't it be $\hat{G}$ in the equation? Also, I think it would be correct to speak of a standard error, not standard deviation (Fig. 5, and L154).

- L167: how were the values for $phi\_0$ and $M\_0$ determined?

- Table 1: a statistical evaluation would be helpful: what is the correlation, the RMSE, the bias of the model?

- Fig. 6/Sect. 3.1: instead of using the interpolated ice thicknesses for evaluation, the profiles should be used. Only this will allow a quantitative and robust assessment of the model results (e.g., what is the correlation between observations and model results? What is the RMSE? Is there a bias?).

- L224-225: the description of the lines should be in the caption of Fig. 9, not in the text.

- Sect. 4.4: see general comment above, but additionally: the selection of studies you compare to that project glacier evolution for individual glaciers seems a bit random. I would suggest to focus only on these that include glaciers in the Southern Andes.

---

## Author Response (AR1)

**Changes made in response to the reviews of the manuscript "21st century fate of the Mocho-Choshuenco ice cap in southern Chile" (tc2020-296)**

**Matthias Scheiter, Marius Schaefer, Eduardo Flandez, Deniz Bozkurt, Ralf Greve**

**May 2021**

In the following, we give an account of the most significant changes that were made in comparison to the initially submitted manuscript, the details of which can be found in the attached latexdiff-file.

In section 1, we outline the changes according to three different categories: changes in the methodology and acquisition of the results; changes in the figures; and changes in the text and the division of sections.

In section 2, we attach a copy of our original responses to the referee comments (including figures from that document). We have followed these responses in most cases, but include an update in blue colour where we have deviated from the original plan.

**1 Significant changes made to the manuscript**

**1.1 Changes in methodology and results**

- The most important change is the replacement of a steady-state spin-up in favour of a transient spin-up in which the model is forced for 35 years before matching the 2013 state. The most important consequences of this are significantly higher ice loss rates until the end of the 21st century.
- In light of the new spin-up, we have re-calibrated the model and are now able to reproduce the negative surface mass balance (SMB) at the stakes which was not the case before. Several mass balance parameters have been adjusted to better match observational constraints.
- We have added a control run for comparison with the future projections, where we force the model with zero temperature anomaly with respect to 2006-2020. This enables us to estimate a committed mass loss and to separate stand-alone ice dynamic effects from the impact of the future projections.

**1.2 Changes in figures**

- Figure 3: We have changed the angle explaining the parameter  $\varphi_0$  from 30° to 315°, which is in line with the value used in the paper.
- Figure 4: We have added the ERA5 temperature trend between 1980 and 2013, which is used for the spin-up.
- Figure 6: We have added three subplots spatial distribution of SMB; evaluation of observed versus modelled SMB at stake locations; evaluation of observed versus modelled ice thickness along the radar lines. We have removed the subplot showing the build-up of the ice cap in the steady-state spin-up, which is outdated in the new methodology. We have also adjusted the order of the subplots to match the order in which we explain them in the text.

- Figure 7 (previously Figure 8): We have added the volume evolution from 1980-2020 in order to show the build-up of the steady-state and the transition to the future projections. We have also added the control run to the plot.
- Figure 8 (previously Figure 9): We have added the volume evolution from 1980-2020 in this plot as well.
- Figure 9 was Figure 7 before, changed due to the restructuring of the sections (see below).

**1.3 Changes in text and section division**

- Section 2.4: New section, included to explain the rationale behind the spin-up and calibration, compare an estimated turnover time of the ice cap to the period of the spin-up, and introduce the ERA5 temperature data.
- Section 3.1: Updated according to the new Figure 6 and the new spin-up.
- Section 3.2: Combines the previous sections 3.2, 3.3 and 3.4, combined to provide a more coherent description of the future projection results. Changed the order and placed the future thickness evolution at the end (hence the change in figure numbers above).
- Section 4.1: Updated according to the new spin-up. Included a comprehensive explanation of the used SMB parameters; included a paragraph justifying the use of a transient rather than steady-state spin-up.
- Section 4.2: Updated according to the new future projections.
- Section 4.4: Changed the order of a few paragraphs we now start discussing other future projections in the Wet Andes, and discuss how they compare (or do not compare) to our results. We then move to other projections with ice flow models in the Andes, and end with ice flow projections world-wide. We discarded a few comparisons which seemed irrelevant, and go more into depth with the global future projections made by Hock et al. (2019) and Marzeion et al. (2020).
- Section 5: Changed name from 'Conclusions' to 'Conclusions & Outlook'; extended the text to give a more detailed summary of our results, and a more detailed account of potential targets for studies in the future.

**2 Original author replies to the referees and deviations from them**

We thank both anonymous referees for taking the time to evaluate our manuscript. In the following, we reply to the comments of both referees and outline how we will incorporate them in the revised version of the manuscript. The figures mentioned in the text are included at the bottom of this document.

**2.1 Anonymous Referee #1**

**2.1.1 Major comments**

**2.1.1.1 Calibration procedure and surface-mass-balance (SMB) scheme**

My first concern is that the parameter calibration isn't adequately evaluated against available mass balance observations. I would argue that matching the overall volume and/or extent is not necessarily a sufficient constraint to conclude the model captures the most important features of the present-day glacier (as is concluded at several points). Given that there are mass balance observations available (already used to calibrate the sensitivity to future warming), there is yet more information that could be used to evaluate the calibrated model parameters and the initial glacier state that they yield.

**Response:** We recognize that the lack of taking into account observed surface mass balance in the calibration of the model was a major inconsistency in our study, and thank the reviewer for pointing this out. We have carefully re-calibrated the model parameters, and attached Figure 1 shows the new fit of observed against modelled annual mean SMB at the stake locations, which we believe shows a good agreement between both.

In particular, the calibrated value for maximum snowfall (S0 = 1.07 m/yr) strikes me as surprisingly low given the high precipitation and mass turnover rates discussed earlier in the paper. The mass balance gradient (line 167) is fixed at a (quite high) value of M0 =  $0.023 \text{ yr}^{-1}$  (2.3 m w.e./yr per 100m vertical?). How was this chosen?

**Response:** As described in the manuscript, these values were chosen in order to match the observed and simulated ice thickness distributions. After re-evaluation, we changed the mass balance gradient M0 to  $0.027 \text{ yr}^{-1}$  which better reflects the true observed gradient, and S0 to 2.2 m/yr which better reflects the SMB in the vicinity of the summit. The denomination of S0 as maximum snowfall might be misleading here and is a remnant from previous applications of SICOPOLIS to the ice sheets, where surface melt in higher elevations is negligible. In our case, calling it maximum SMB is more reasonable and in order to avoid confusion we will change the name in the final version of the manuscript.

One thing that strikes me from these values of S0 and M0 is that mass balance must plateau very quickly (< 50 vertical meters) above the ELA, implying a large area where mass balance is uniform at the maximum value. Is this backed up by the available mass balance data? From my own look at Schaefer et al., (2017), it appears that both accumulation and ablation are substantial all the way to the summit in the seasonal balances (e.g., their Fig. 7). If there is in fact non-zero ablation over the entire glacier surface, the entire surface should be susceptible to surface-mass-balance anomalies caused by further melt-season warming. However, if I understand the existing SMB scheme correctly, the model assumes warming doesn't actually cause a SMB anomaly in grid cells above the maximum snowfall cutoff. And, as noted above, this seems like a large initial area, due to the values of S0 and M0. Should warming really only affect the lower reaches of the glacier?

**Response:** We thank the reviewer for this sharp observation of the impact the chosen parameters had on the distribution of SMB over the ice cap. The parameters of the re-calibration fix these issues, and Figure 2 shows the SMB map resulting from the new parameters. The patterns here are in good agreement with the observed SMB map (Figure 9 in Schaefer et al., 2017).

Obviously, no simulation can be expected to capture every detail of the mass balance, and I think a simple elevation/aspect dependent scheme can be a reasonable approach. But as described it seems

like there are some embedded assumptions that may not be consistent with observations. The pattern of mass balance and the pattern of anomalies driven by warming should be include in the evaluation of optimal model parameters, given that stake data are available. If I have misinterpreted the SMB scheme, please clarify. A figure showing the initial spatial pattern of SMB could really aid the reader in interpreting how the maximum snowfall and aspect-dependent ELA affect SMB on the actual topography (e.g., beyond the schematic in Fig. 3).

**Response:** We think that our re-calibration addresses most of the reviewer's concerns and, within the limitations of a strongly simplified SMB scheme, leads to a better representation of SMB in our model. This becomes evident in Figure 1, showing the performance of the model at individual stakes, and Figure 2, showing the overall SMB pattern on the ice cap. We propose to include Figure 2 of this document as a subplot in Figure 6 of the manuscript, and to put Figure 1 in the supplementary material. Eventually, we also included Figure 2 as a subplot of Figure 6 rather than the supplement, as it is an important part of our spin-up evaluation.

**2.1.1.2 Assumption of steady-state for spinup**

My second major comment is on the steady-state spinup for the initial condition. Assuming a steady state is questionable, given more than a century of global warming over the industrial era, to which glaciers respond with a dynamical lag. This lag means that glaciers, in general, are out of equilibrium with current climate (see, e.g., Lüthi et al., 2010; Christian et al., 2018; Marzeion et al., 2018; many others). Forcing a steady state can throw off parameters in the initial calibration (e.g., the ELA), and could throw off the initial transient response when forcing is applied.

The observations of negative mass balance (noted on line 58; from Rivera et al., 2005 and Shaefer et al., 2017) are themselves an indication of disequilibrium, and another reason to include SMB in the model calibration (granted, 5 years is not many observations to define a mean balance). The Rivera et al. study also shows substantial retreat since at least 1976, which would also suggest that the early 2000s extent is not likely to be a steady state.

At the very least, I think it is necessary to estimate how far from steady state the ice cap is at the time the simulation starts, in case the projections need to be qualified in light of this assumption, or the model recalibrated. Some first-order estimates could be made based on the ice cap's estimated response time (e.g.,  $H/b_{terminus}$ , see Johannesson et al., 1989). If the response time is long and disequilibrium substantial, it would be necessary to start the simulation earlier to properly capture the transient response for future projections. This is especially true for evaluating the difference between RCP 2.6 and 8.5 trajectories, as the "committed" response to past warming may be a substantial part of the true 2.6 trajectory (with little additional warming), but this would not be captured if the model starts in a steady state in the 2000s.

**Response:** We thank the reviewer for their comments regarding the appropriateness of our steadystate assumption. According to the formula indicated by Johannesson et al. (1989), we compute a response time of approximately 37 years (taking a maximum measured ice thickness of 261 meters and a minimum yearly SMB of -7 m w.e./yr at stake B9, close to the terminus). As this response time is relatively high, we have replaced the steady-state spin-up by a transient spin-up that takes into account ERA5 temperature data between 1979 and 2013, a 35 year period similar to the response time. Given a trend temperature increase of around 0.2 K over this period according to ERA5, we first create a steady-state with an anomaly of -0.2 K with respect to 2013. Then, we force the model with the temperature evolution over 35 years until 2013. The area and volume both match the observations around 2013 well. This out-of-balance glacier state is then forced by the different future scenarios in order to yield projections until 2100. As expected, the future ice loss is now significantly more pronounced than previously, as the projections start off with a negative slope that was not present in our previous projections (see Figure 3). As the new transient spin-up reflects well both SMB and geometry of the current ice cap, we are optimistic that our new projections are a reasonable guess of future ice cap evolution under the applied temperature projections. We ended up calculating the turnover time according to another equation which is conceptually very similar to that of Johannesson et al. (1989), giving a time of 27 years and therefore underpinning the conclusion made about the viability of a 35-year transition period. Details in section 2.4 of the new version of the manuscript.

**2.1.2 Minor comments (line by line)**

- Line 23: change 10000 mm/yr to 10 m/yr ? (That is a lot of zeros for the readers' eye!)

**Response:** We will make this change in the manuscript.

- 27: "As these are best represented in ice-flow models, they are. . ." Clarify wording: "these" and "they" presumably refer to different things here, but sentence is ambiguous

**Response:** The sentence will be changed to "Ice-flow models incorporate these processes, and are therefore appropriate tools to project the future behaviour of the glaciers of the Wet Andes."

- 59-61: how are you defining mass turnover here? Can you elaborate on how these temperature measurements indicate this?

**Response:** We will clarify in the text that the rather high precipitation rates lead to high accumulation rates, and that the high mean temperature leads to high melt rates, together resulting in a high mass turnover.

- 87: "we replaced solving the energy balance by.." somewhat awkward wording, consider rephrasing

**Response:** Will be changed to "...we do not solve the energy balance equation. Rather, we keep the temperature..."

- 100: "SMB should be lower"... is it? Based on observations or simulations? It would be helpful to discuss what processes likely lead to this pattern, to help the reader understand how much the model may be simplifying reality.

**Response:** We will clarify that SMB should be lower in the north-west mainly due to the influence of solar radiation and wind-redistribution, as indicated by observations.

- 100–104 and Fig. 3b: I'd suggest making the angle in the schematic correspond to the angle used in the actual simulations. I initially got confused as the wording in the text (referring to Mocho) doesn't correspond to the orientation in the schematic.

**Response:** The angle in the schematic will be changed.

- 107: if the ELA is defined by the angle with respect to the summit, and the mass balance gradient is constant, doesn't this lead to very sharp spatial variations in mass balance as points near the summit? I suppose the maximum snowfall could limit this, but is this a realistic pattern? Again, a spatial map of SMB could be useful for the reader.

**Response:** This would be the case for very high values of S0, but not for the range of values realistic for our ice cap. (see attached Figure 2)

- 119: Clarify: model mean, time mean or both?

**Response:** Sentence will be changed to: "For each individual model, the mean temperature between 2006 and 2020 was then subtracted from the whole time series, leading to anomaly temperature projections with respect to this period."

- 136: Is precipitation taken into account for this regression? That will affect ELA variability too... It is one thing to only consider temperature for the future projections (but see later comment), but I would think the effect of each year's accumulation should be taken into account to calibrate this relationship, especially with only 4 years available.

**Response:** We will clarify our assumption that only temperature has influence on future SMB, and only on the parameter describing the mean ELA  $(B\_ELA)$ .

- 136: Also, is there a particular reason the temp-ELA relationship and future projections are based on annual-mean rather than melt-season temperatures? Do the climate models predict melt-season temps warm at the same rate as annual mean?

**Response:** We notice that both annual and melt season (DJF) temperature projections of the climate models are very close to each other for the Mocho-Choshuenco volcano, indicating no important differences in the warming rates, which can give us the confidence to focus on annual scale. We will add a note on this in the revised version.

- 167: Again, where are the values for M0 and phi taken from?

**Response:** We will change this paragraph to take into account the new SMB calibration, and will clarify that we chose  $\varphi_0 = 315^{\circ}$  due to wind redistribution and solar radiation, and obtained  $M_0 = 0.027 \text{ yr}^{-1}$  as observed mass balance gradient from the stakes. Instead of changing the paragraph, we have included the new section 2.4 that explains the spin-up, and provide a detailed analysis of the SMB parameter values in section 4.1.

- 172: "does not reproduce well" ... Consider rewording for clarity.

**Response:** Will be changed to "... where the simulation is not in a good agreement with the observations". The whole section has changed, so this unclear formulation is not present anymore.

- 177: 200 m/yr is very fast! Can you comment on why the model might give velocities an order of magnitude higher here than most values in Table 1? Is it the geometry, or SMB pattern that allow this?

**Response:** In our new spin-up, the ice tongue is much shorter and does not go beyond the observed glacier outlines anymore. The velocities around and below the stake B12 are maximum 60 m/yr now, which is similar to the observed velocity of B12. As the SMB on the ice cap is the main difference between the original version of the manuscript and this update, we assume that the high velocities were caused by the previous SMB parameterization.

- 181: I'm just curious if you know why there are seasonal but not annual velocities? I'm surprised annuals weren't derived from, e.g., mass balance stake locations. Were seasonal velocities only measured in one year?

**Response:** Currently, the only available observations on ice flow velocity at the surface are the seasonal velocities from Geoestudios (2013), as mentioned in Section 2.1.

- 183: As I understand from the SMB parameterization, precipitation is implicitly assumed to not change. Is this roughly consistent with the model projections and/or observed trends for the area? I'd expect that temperature is the main forcing, but recommend at least stating that this assumption is made.

**Response:** The parameterization is only for the net SMB; it does not explicitly distinguish between precipitation and runoff. We assume that SMB changes correlate with surface temperature changes. Of all SMB parameters only the mean ELA (B\_ELA) is changed according to temperature (and not the other parameters such as S0, M0, PHI0, and A\_ELA). We will clarify these assumptions in the paper. Clarified in the beginning of section 2.6.

- 187: Retreat is strongest in the north for RCP 2.6... is this partly because of the imposed higher ELA and cap on mass balance?

**Response:** Our new spin-up has a different thickness distribution in the north (where thickness is not well known due to sparsity of observations), so the description and interpretation of these results will change in the final version of the manuscript.

- 215–17: I find this statement on internal variability confusing. Do you just mean that the spread

due to different climate models and scenarios hasn't had time to diverge? Consider rewording for clarity.

**Response:** Yes, that is what we meant. We will clarify this statement, even though the observed effect is less pronounced in the new results.

- 220: When considering a different ELA-temperature relationship, doesn't this imply other parameters are also different (e.g. the vertical SMB gradient?). Does the initial state reflect differences in these parameters, if any?

**Response:** The mean ELA is the only parameter that is being influenced by the increasing temperatures. Due to lack of observational data, there is no clear indication of how the SMB gradient would change with temperature, and we therefore leave it constant at the value obtained from calibration.

- 238: "high observed velocities" ... do you mean high modeled velocities (referring to an area without extant ice)

**Response:** Yes, this was an unclear formulation. We meant "observed in the modelling results", but will make this more clear. The very high velocities are not present in the new results, and this part of the discussion has changed.

- 248: I think you mean thinner here, right?

**Response:** Correct. We will change this.

- 246–48: Here you have explained the low velocities in terms of anomalously thin ice, but why is the ice too thin? The combination of too-thin and too-slow together indicate that overall fluxes are underestimated in these areas... which ultimately seems like a mass balance issue. Could this be related to the rather low cap on mass balance (see major comment above)?

**Response:** After changing the SMB parameterization, the ice on the south-western part of the ice cap is still too thin in the simulations, and the velocities too low. As the new SMB parameterization reproduces the observed SMB well (see Figures 1 and 2), we assume that these inconsistencies of our model are not primarily an SMB effect. We rather suspect dynamical reasons, and will add this in the discussion.

-253–54: I find it a bit circular to invoke a "stable position at the moment" to explain a model result, when the stable position is imposed by your choice of a steady-state initial condition. This is one area where the steady-state assumption (see major comment) can affect projections.

**Response:** These parts of the discussion will change according to our new projections which have changed significantly (see Figure 3).

- 266: Suggest word choice other than "unstable". . . there's no instability in a dynamical sense here, just a larger forcing.

**Response:** We will exchange lines 265-270 by: "The high-end atmospheric warming scenario RCP8.5 causes a highly accelerated ice loss from the 2040s to the 2080s with high retreat rates, before becoming more subtle from 2080 to 2100, which can be explained by the fact that most ice has already melted away." We didn't use this sentence, but changed the discussion in the spirit of the reviewer comment.

- 316: "would have disappeared" » projected to disappear?

**Response:** We will change this formulation.

- 319: "maintained glacier area constant" » "maintained a constant glacier area" ?

**Response:** Will be changed.

- 330: "lost majority of their ice mass" ... relative to preindustrial?

Response: We will add "... relative to the glaciers observed at present".

- 331: "The" » this

**Response:** Will be changed.

- 331-332: general comment here that different proportions of volume lost over a given timeframe can be due simply to different glacier geometries/hypsometries. I think that should be borne in mind for all of the comparisons in this section...

**Response:** We will mention the impact of glacier geometry on volume loss in this section in the final version of the manuscript.

- 357: missing period

**Response:** Will be added.

- 370: then  $\gg$  than

**Response:** Will be changed.

**2.2 Anonymous Referee #2**

**2.2.1 General/major comments**

**2.2.1.1 Assumption of steady-state for spin-up**

[T]he paper starts by introducing the glaciers of the southern Andes having among the highest mass losses of all glacier regions worldwide, and specifies a SMB of almost -1 m w.e./year in observations for the Mocho-Choshuenco ice cap (L58). However, the method assumes a zero mass balance of ice cap under present-day (2006-2020) conditions, by requiring that the ice cap's geometry is closely reproduced by the model as a steady state at a temperature anomaly of zero. This is a strong internal inconsistency which is currently not at all addressed in the paper.

**Response:** We thank the reviewer for pointing out the inappropriateness of a steady-state as starting point for the future projections. We have addressed these concerns by creating a transient rather than steady-state spin-up. Taking into account 35 years of ERA5 temperature data (1979-2013), we first create a theoretical steady-state for the 1970s and then force the model with the ERA5 data until 2013. This spin-up leads to an ice cap in an out-of-equilibrium state in 2013, and we hope it satisfies the concerns of the reviewer. The subsequent future projections are much more negative than before, using a steady-state spin-up for 2013 (comparison between Figure 3 below and Figure 8 in the original manuscript).

Closely related is the lack of discussion of the parameter values obtained by matching the steady-state thickness as closely as possible to observations: equation 9 indicates  $B_ELA = 1777$  m from the observation as opposed to 2035 m from the observation, which (again according to eq. 9) corresponds to a temperature offset of almost 3 K ((2035 m-1777 m)/88m/K).

**Response:** There seems to be a misunderstanding here, as we do not state that  $B\_ELA = 1777$  m. The value of 1777 m is rather the intercept of the weighted linear regression we perform in this section. It is therefore the value the ELA would take if the mean annual temperature was 0°C, as indicated in line 140. With a temperature-ELA gradient of 88m/K and a mean annual temperature at the ice cap of around 2.6°C, this regression leads to a mean ELA of 1777 m +  $88m/K \cdot 2.6K \approx 2005$  m, which corresponds the observed mean ELA. This section is not related to the calibration

of  $A\_ELA$  or  $B\_ELA$ . In order to avoid future confusion, we will clarify the meaning of the 1777 m by stating it after equation 9, and will in the opening paragraph clearly state that this section is only about observations and focused on finding a relationship between temperature and ELA, and not aimed at calibrating the model parameters.

The turnover of 5 m w.e./year (L59) is in apparent contradiction to a maximum annual snow fall S 0 of about 1 m as best parameter values (L166).

**Response:** Instead of referring to a mass turnover of 5 m w.e./yr, we will state a high mass turnover due to a mean modelled accumulation of 3.5 m w.eq. (see Figure 8 in Schaefer et al 2017). Also, we recognise that the denomination of S0 as maximum snowfall is misleading in the context of this study, and will change the name to "maximum SMB" in the final manuscript. The name is a remnant from previous studies involving SICOPOLIS. The new value of S0 2.2 m w.e./yr is closer to the observations. We have changed the value of the mass turnover to 2.6 m w.e./yr, as calculated in section 2.4.

The initialization of an ice flow model is a complex task, but has been addressed before (e.g., Eis et al. 2019, DOI: 10.5194/tc-13-3317-2019; Zekollari et al. 2019, DOI: 10.5194/tc-13-1125-2019). These studies may be helpful for coming up with an adequate initialization approach.

**Response:** We thank the reviewer for these literature suggestions, which were indeed helpful in producing a transient spin-up for the year 2013. Due to different data availability and study scope, we did not implement these exactly, but our study follows the main idea of both initialization procedures.

**2.2.1.2 Lack of validation of modeled SMB**

[T]he authors chose the somewhat unusual way to calibrate parameters of the mass balance equation through matching observed and modeled ice thickness, which I would assume are closer related to parameters of the ice flow model (which are also included in the observation). I don't understand the rationale of this approach, given that mass balance observations are available, and could easily be used for optimizing the mass balance parameters. At the same time, an evaluation of the model's performance concerning SMB is completely lacking.

**Response:** Our previous calibration of SMB parameters and sliding parameter was focused on matching the geometry of the ice cap by minimising the RMSE of modelled against observed ice thickness, and out of the best models choosing the one that best matches the observed ice volume, as explained in lines 162-167. This procedure was successful as it was able to reproduce the overall geometry of the ice cap, in terms of volume, mean ice thickness, and area (but, as pointed out by both reviewers, lacking an evaluation of SMB). We disagree with the statement that matching the geometry is an unusual approach, as many studies have done this in the past, including the two cited above by the reviewer (Eis et al., 2019; Zekollari et al., 2019). We agree that it is important to also validate the modelled SMB against observed SMB, which was lacking in our previous approach. We thank the reviewer for pointing this out, and have carefully re-calibrated the SMB parameters. The new parameters are Phi  $0 = 315^{\circ}$ , A ELA=87.5m, B ELA=2050m, M  $0 = 0.027 \ 1/yr$ , and S  $0 = 2.2 \ m \ w.e./yr$ . This parameterization matches better both the observed SMB at the stakes (see attached Figure 1) and the SMB distribution of the ice cap (see Figure 2). We are therefore confident that the SMB in our model reflects the SMB of the real ice cap better than previous to the reviews. We suggest to include Figure 2 of this document in the main paper, and Figure 1 in the supplementary material. We have included Figure 1 as part of Figure 6 instead of putting it in the supplement, as these results are essential for the spin-up evaluation.

Since a steady state condition is used for the recent past as spin-up, and observations of the SMB during a very similar period are 0.9 m w.e./yr, I suspect that the model has a positive bias of around +0.9 m w.e./yr (not exactly, because the observations only cover a fraction of the glacier's surface). If these presumptions are correct, this would imply that also the projections have a strong positive SMB bias, such that they would strongly underestimate the future rate of mass loss. It is good that the authors evaluate their results against ice thickness and velocity observations, but with the application in projections of mass change in mind, the evaluation of the SMB results is even more important. Without a convincing evaluation the projections cannot be trusted.

**Response:** We thank the reviewer for raising these concerns. Indeed, from our experiments it appears to be the case that the transient spin-up imposes a more negative mass balance on the ice cap than a comparable steady-state spin-up. As mentioned above, we have updated our SMB calibration approach, and we hope that the attached Figures 1 and 2 provide the evaluation the reviewer asked for.

**2.2.1.3 Discussion of comparable studies**

At two occasions in the manuscript (L29ff L312ff, the authors state that there are few studies that have projected the future evolution of glaciers in den Andes). Among the studies they cite is Hock et al. (2019), which alone summarizes six studies; a more recent intercomparison is Marzeion et al. (2020, DOI: 10.1029/2019EF001470), which includes seven different models. These are additional to the ones discussed in the paper, but very different in that they don't focus on one (or a few) individual glacier(s), but include all glaciers worldwide. I think it is possible to turn this study into a publishable paper even though by now, there are many models around that have been applied to this specific glacier. But it will be necessary to go into the individual model publications (not just the intercomparison paper, as done now) and see how they are approaching the problem, and discuss the merits of the approach used here approach in this context: what are the advantages of their mass balance parameterization over those used in the models summarized in Hock et al. (2019) and Marzeion et al. (2020)? What are the advantages of using SICOPOLIS instead of the (mostly simpler) approaches in the global models?

**Response:** We agree with the reviewer that the mentioned global studies are valuable as they compare all or most glaciers world-wide under similar conditions and derive conclusions on the general state and future evolution of glaciers, and important consequences such as sea-level rise and fresh-water supply. However, it is of at least equal importance to perform local, high-resolution studies, both to inform in detail about the specific conditions in the area, and to give an estimate against which the results of global studies can be evaluated. The simplifications of most global studies lead to inconsistencies on a local scale. For example, simple area-volume relationships (averaged over a huge amount of glaciers) do not account for the local specifics. Simple SMB parameterizations (as in most studies of Hock et al. (2019) and Marzeion et al. (2020)) do not take into account small-scale variations such as introduced by our aspect-dependent parameterization, but likely treat the whole ice cap as one grid point. On the Mocho-Choshuenco ice cap, a relatively large amount of small-scale, high-resolution observations is available (more than on most other glaciers in the southern Andes), and it therefore gives a great opportunity to perform a high-resolution study against which studies of coarser resolution can be compared. Another concern about the studies featuring in the intercomparison studies is that most of them do not include a proper parameterization for frontal ablation and calving (as already mentioned in our discussion in lines 333-340). As this is an important form of mass loss in Patagonia, and Patagonian glaciers in turn are dominant in analyses of the whole southern Andes, the conclusions possible to draw on the Mocho-Choshuenco ice cap are likely limited. In such cases, our study (and many other local/regional studies performed world-wide) can provide useful feedback on the performance of global modelling approaches. We are therefore convinced that our study is valuable (even more so after the considerable improvements that were made based on the reviewers' suggestions) and makes an important contribution to the understanding of glacier behaviour in the southern Andes.

I am convinced that once the first two major issues are addressed by the authors, the results will change substantially. I have therefore abstained from providing detailed/minor comments to the sections that present or discuss these results. These should be addressed at a later stage, if the authors decide to revise and resubmit the paper.

**Response:** We have addressed both major issues in our new SMB calibration and the transient spinup. The results have changed and we will update the sections that present and discuss them, and are happy to receive further comments on them afterwards.

**2.2.2 Specific/minor comments/suggestions**

- L14: I don't see this generalization backed up by the study results.

**Response:** Most glaciers in the area have a similar setting, both in a geographical and climatological sense. We provide the first projections based on ice-flow modelling for this area, which gives a volume loss up to 94% until the end of the century. We think it is reasonable to expect that if this ice cap loses almost all its mass the neighbouring glaciers would be affected to a similar degree.

- L24: I assume there is a strong seasonality in this number; it would be helpful to be a bit more specific.

**Response:** We will add that the cited reference states an overall modest seasonality for the Wet Andes.

- L54: repetitive, can be shortened.

**Response:** We will change the first two sentences of this paragraph to "The Mocho-Choshuenco ice cap covers the Mocho-Choshuenco volcanic complex, which is located in the Chilean Lake District at ...."

- L61: it can be explained by the climatology, as documented in the data – not the data itself.

**Response:** We will change this formulation.

- L62: based on the setting of the station and the glacier (orography, wind direction, etc.) would you expect precipitation at the glacier to be higher or lower than in Puerto Fuy? A qualitative assessment would be helpful for readers unfamiliar with the area.

**Response:** We will add in the final version of the manuscript that orographic precipitation effects lead to a considerably higher amount of precipitation on the ice cap than in Puerto Fuy.

- L76: is an uncertainty assessment available either the total volume?

**Response:** We assume that the reviewer refers to line 66. Unfortunately, the study we cite regarding the estimated volume of the ice cap does not provide an uncertainty assessment.

- Fig. 3b and discussion around it is a bit confusing. Assuming that A\_ELA and B\_ELA are both positive, and that y is latitude, the maximum ELA would be in the north-east sector. However, the text says the SMB should be lower (equivalent to a higher ELA) in the north-western sector. Since you prescribe phi\_0 anyway (L167), why not make Fig. 3b using the actual values used?

**Response:** We will change the schematic and display the angle phi\_0 in a north-western direction.

- L136: by the ELA, M\_0 and S\_0.

**Response:** We will clarify our assumption that future temperature anomalies only affect the mean ELA of the ice cap  $(B\_ELA)$ .

- Eq. 5 and following: I'm unfamiliar with this notation. Please explicitly define N. Shouldn't it be Ghat in the equation? Also, I think it would be correct to speak of a standard error, not standard deviation (Fig. 5, and L154).

**Response:** We will change "bivariate normal distribution" in line 143 into "bivariate normal distribution  $\mathcal{N}(\mu, \Sigma)$ ", and G into  $\hat{G}$  in equation 5.

- L167: how were the values for phi 0 and M 0 determined?

**Response:** We will change this paragraph to account for the new SMB calibration, and will clarify that  $M_0$  is fixed to represent the observed SMB gradient, and phi\_0 is set to 315° in agreement with the observed effects solar radiation and wind-redistribution have on the ice cap. The new spin-up and calibration are explained in section 2.4, and the values of the SMB parameterization are discussed in

section 4.1.

- Table 1: a statistical evaluation would be helpful: what is the correlation, the RMSE, the bias of the model?

**Response:** We think that a statistical evaluation of as few as six value pairs has to be treated with caution, especially given the fact that we are here comparing seasonal against annual velocities, and just aim to get an overall insight of the magnitude of the velocities. This is given in the current form of the table. However, we will indicate an RSME of 12.5 m/yr and an average underestimation of 9.4 m/yr of the simulated values compared to the observed ones.

- Fig. 6/Sect. 3.1: instead of using the interpolated ice thicknesses for evaluation, the profiles should be used. Only this will allow a quantitative and robust assessment of the model results (e.g., what is the correlation between observations and model results? What is the RMSE? Is there a bias?).

**Response:** In the attached Figure 4, we plotted the observed ice thickness values along the profiles against the simulated ice thickness on the same location (obtained through interpolation from the grid to the profile points). It shows an overall good fit with a high correlation coefficient and a low RMSE compared to the magnitude of the values. On average, the simulated ice thickness is around 10 m lower than the observed ice thickness on the profiles. When performing a similar analysis on the gridded data (interpolated observations against modelled ice thickness), the values are similar, but the bias is lower, with the model underestimating ice thickness by around 2 m on average. We will add this information in the manuscript, and Figure 4 in the supplementary material. We have included this figure as part of Figure 6, as it is essential for the evaluation of the spin-up.

- L224-225: the description of the lines should be in the caption of Fig. 9, not in the text.

**Response:** We will put the description into the figure caption.

- Sect. 4.4: see general comment above, but additionally: the selection of studies you compare to that project glacier evolution for individual glaciers seems a bit random. I would suggest to focus only on these that include glaciers in the Southern Andes.

**Response:** In this section, we aim to put our study in context with other studies that have performed ice-flow simulations. The only other ice-flow simulation that projects future glacier behaviour in the Andes under climate change scenarios is that of Réveillet et al. (2015) on Zongo glacier in the tropical Andes, and there are no studies available for the southern Andes. In the absence of direct comparisons, we think it is valuable to compare the ice loss projected by our model to the losses projected in other areas in the world with similar methodologies. However, we agree that it can also be helpful to go into more detail with the comparison studies, and will do so in the final version of the paper.

Figure 1: Comparison between modelled and observed surface mass balance at the stake locations after re-calibrating the SMB parameters. This figure was slightly altered and is now Figure 6f in the new version of the manuscript.

---

## Author Response (AR2)

**Authors' reply to second round of referee comments on "21st century fate of the Mocho-Choshuenco ice cap in southern Chile" (tc2020-296)**

Matthias Scheiter, Marius Schaefer, Eduardo Flandez, Deniz Bozkurt, Ralf Greve

June 2021

*In the following, we address the minor revisions of the anonymous referee that provided feedback to our revised manuscript. We thank the referee for taking the time to carefully review our manuscript a second time and for providing another set of helpful comments. Apart from the changes outlined here, only minor additional corrections were made, as can be seen in the 'track-changes file'.*

**1  Anonymous Referee #1**

I think this is a very and thorough revision of the study and manuscript. I see that the major changes are an updated surface mass balance parameterization and also a transient spin up to capture the ice cap's modern disequilibrium. Thanks for taking the time to carefully address these points (which I note the other reviewer raised as well), and nice work integrating them.
The adjusted methods and the reasoning behind them are described well, and I think the new figure panels will help the reader evaluate the model calibration and spinup. Based on these figures/analyses and the updated projections, I agree with you that the model seems to better capture the recent state of the ice cap – and that lends more confidence to the projections
I have just a few remaining comments, which are mostly minor questions of clarity or wording. Hopefully these will be quick to address.
I support publication of the revised manuscript. I think it will make a nice contribution both as a case study for glacier change in this region, and a useful demonstration of applying a 3D ice-flow model to a mountain glacier/ice cap setting.

**Response:** *We thank the reviewer for the positive feedback and are happy that our changes seem to have satisfied their previous criticisms.*

7: "exposition" – consider changing to aspect, which I think is more familiar and also used later in the manuscript

**Response:** *Changed.*

26-27: "apart from ice dynamics..." – consider re-wording; somewhat confusing to separate these with "apart from", as ice dynamics relate the climate forcings to glacier change

**Response:** *Sentence changed to 'In addition to climate forcings, other important contributors to glacier change in the region are ice dynamics and frontal ablation'.*

97: I assume the altitude is the (evolving) ice surface elevation, but perhaps you could specify

**Response:** *'Altitude' changed to 'evolving ice surface elevation'.*

134: I think estimating the local warming since 1979 based on both radiosonde and reanalysis is a good approach. This isn't a criticism, more of a comment: I was just surprised to see it is only a  0.2 C warming – much less than the global average over this time. Could be worth just pointing that out in case readers have other regions with stronger recent warming in their minds.

**Response:** *Clarified by changing 'long-term temperature trend' to 'long-term regional temperature trend'.*

136-37: I'm a little confused by this description – shouldn't the ELA be lowered to make the 1979 steady state?

**Response:** *'elevating' changed to 'lowering'.*

152: model » models

**Response:** *Changed.*

176: "of availability of. . . " – suggest changing to "where both the ELA and. . . are available"

**Response:** *Changed.*

Fig. 6: I assume this is for the present day (transient) state, not 1979? Consider stating to clarify.

**Response:** *Has been clarified by stating that it is the ice cap state of 2013.*

224: suggest "until 2100" » "by 2100"

**Response:** *Changed.*

256: "ice loss accelerates" – this is thickness change, right? I think I understand what you mean, but consider clarifying as this seems at odds with earlier statement about how the mean curves flatten by 2100 (229).

**Response:** *Clarified by changing 'ice loss clearly accelerates in the second half of the century' to 'ice loss clearly accelerates between 2040 and 2080'.*

309: "dynamical processes" – consider wording to "dynamical state"? Or transient state? The spinup to me seems more about capturing the state rather than new processes.

**Response:** *Clarified by changing 'the calculated times indicate that the most important dynamical processes of the ice cap are found to be captured by our model' to 'the calculated times indicate that the most important features of the transient state in 2013 should be captured by our model'.*

332: Opposed » as opposed

**Response:** *Changed.*

342: suggest "climate" rather than "circulation" for consistency

**Response:** *Changed.*

364: of advantage » advantageous

**Response:** *Changed.*

370: Drawback » a drawback

**Response:** *Changed.*

378: few » a few (??)

**Response:** *After looking up the difference between 'few' and 'a few', we decided that 'few' conveys our message better and have therefore kept it.*